# HOW RELIABLE IS LANGUAGE MODEL MICRO-BENCHMARKING?

**Gregory Yauney**   **Shahzaib Saqib Warraich**   **Swabha Swayamdipta**
Thomas Lord Department of Computer Science
University of Southern California
Los Angeles, CA, USA
gyauney@gmail.com   {warraich,swabhas}@usc.edu

## ABSTRACT

Micro-benchmarking offers a solution to the often prohibitive time and cost of language model development: evaluate on a very small subset of existing benchmarks. Can these micro-benchmarks, however, rank models *as consistently as* the full benchmarks they replace? And can they rank models *more consistently than* selecting a random subset of data points? In many scenarios, we find that the answer is no. We introduce a meta-evaluation measure for micro-benchmarking which investigates how well a micro-benchmark can rank two models as a function of their performance difference on the full benchmark. This approach can determine which model pairs can be ranked correctly by a micro-benchmark, allowing for a finer-grained analysis of the trade-off between micro-benchmark size and reliability. Prior work has suggested selecting as few as 10 examples; we find that no micro-benchmarking method can *consistently* rank model pairs 3.5 points of accuracy apart on MMLU-Pro or 4 points apart on BIG-bench Hard. In order to consistently rank model pairs with relatively similar performances, we show that *often as many as 250 examples must be selected, at which point random sampling is competitive* with existing micro-benchmarking methods. When comparing only 8B instruction-tuned models on MMLU-Pro micro-benchmarks with 25 examples, we find that more than half of pairwise comparisons are not likely to be preserved. Our work provides actionable guidance for both micro-benchmark users and developers in navigating the trade-off between evaluation efficiency and reliability. Code and data are at: github.com/dill-lab/micro-benchmarking-reliability

## 1 INTRODUCTION

Micro-benchmarking methods reduce the evaluation time of language models by predicting performance on a full benchmark from performance on a small subset (Vivek et al., 2024; Polo et al., 2024; Perlitz et al., 2024; Ye et al., 2023; Gupta et al., 2025). Current micro-benchmarking determines these subsets based on different criteria (§2). For instance, Anchor Points (Vivek et al., 2024) selects the centroids of test example clusters in the space of model predictions. tinyBenchmarks (Polo et al., 2024) selects examples close to the centroid of clusters obtained using Item Response Theory (Cai et al., 2016). But there is a general trade-off between efficiency and the reliability of the judgments drawn from *any* evaluation set: smaller eval sets are more cost-effective, but they do not always accurately reflect which model will perform best in practice (Shalev-Shwartz & Ben-David, 2014; Dror et al., 2018). Do micro-benchmarks suffer from this general trade-off, as well? Specifically, we ask: How can we measure the extent to which micro-benchmarks reflect the model performance judgments of full benchmarks?

Prior work has evaluated micro-benchmarks using two meta-evaluation approaches: (i) how well they reconstruct the accuracy of any single model on the full evaluation set (Polo et al., 2024), and (ii) how well they preserve the aggregate rankings of a set of models (Vivek et al., 2024; Perlitz et al., 2024). We evaluate micro-benchmarks instead on their ability to predict *pairwise model rankings* on the full benchmark: Given that model $M_1$ outperforms model $M_2$ on the full benchmark, what is the probability that model $M_1$ outperforms $M_2$ on the micro-benchmark? Inspired by statistical power analysis (Card et al., 2020), we measure the minimum performance difference between models

$M_1$ and $M_2$ on the full benchmark that still consistently yields a correct pairwise ranking of the two models on the micro-benchmark. We introduce a meta-evaluation measure, the *Minimum Detectable Ability Difference* (MDAD) that offers a fine-grained view of *which* performance judgments are preserved by a micro-benchmark (§3). A direct consequence of MDAD is an understanding of how micro-benchmark size affects its reliability across model pairs being compared (§5). In addressing pairwise estimates, MDAD offers complementary benefits to existing meta-evaluation measures, which deal with pointwise estimates or aggregate rankings (§5.1).

Unlike micro-benchmark selection, selecting examples uniformly at random has the advantages of speed and simplicity: it does not require evaluating models to learn prediction correlations, nor does it train auxiliary models of instance difficulty. However, existing meta-evaluation has not characterized when micro-benchmark selection outperforms random sampling (Vivek et al., 2024; Polo et al., 2024). Our meta-evaluation measure, MDAD, reveals that the intuitive baseline of random sampling is competitive with existing micro-benchmark selection under all but the most extreme dataset reductions (§5.2). We also use MDAD to show the limits of micro-benchmarking in the common setting of comparing same-size models, which often have similar performances on a task. When selecting 25 examples from MMLU-Pro, 51% of pairwise comparisons among a set of 8B-parameter instruction-tuned models are not likely to be preserved (§5.3).

Figure 1 compares our evaluation with others on revealing the limits of micro-benchmarking. When comparing nearly 100 models on MMLU-Pro (Wang et al., 2024), Kendall's tau rank correlation between the full benchmark and a micro-benchmark shows that Anchor Points is better correlated with the full benchmark than random sampling is at extremely small dataset sizes (Figure 1, top left). However, a correlation of, say, 0.74, does not identify *which model comparisons remain challenging* for a micro-benchmark. Our measure, MDAD, considers the probability that a micro-benchmark agrees with the full benchmark *as a function of the accuracy difference between a model pair* (Figure 1, bottom panels). For two models that differ by 2 points of accuracy on the full benchmark (dashed lines in Figure 1, bottom), we show that when only 10 examples are selected, no micro-benchmark can distinguish these models more than 65% of the time (Figure 1, bottom left). In contrast, when 500 examples are selected (Figure 1, bottom right), many micro-benchmarks can distinguish the same models more than 90% of the time, including even random sampling, which Kendall's tau rank correlation corroborates (Figure 1, top right). Thus, if a practitioner wants to distinguish models 2 points of accuracy apart, then they could simply use random sampling to select at least 500 examples. If instead they wanted only to distinguish models that differ by more then 4 points, then only 10 examples selected by Anchor Points would suffice.

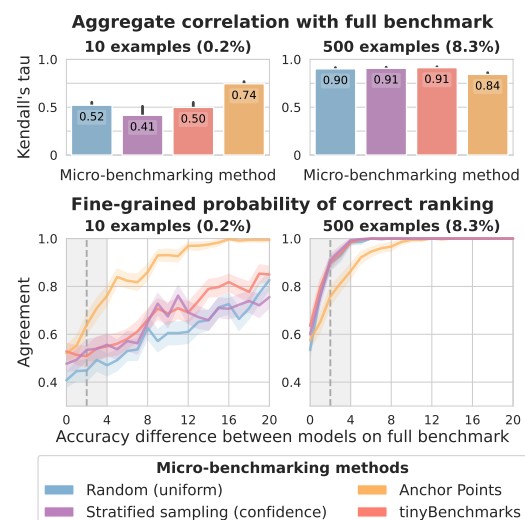

Figure 1: Existing meta-evaluation metrics (e.g. Kendall's tau rank correlation) summarize micro-benchmark performance for MMLU-Pro in the aggregate (top). At extreme dataset reductions, micro-benchmarks can yield high aggregate rank correlation with full benchmarks (top left), but no micro-benchmark has a high probability of agreeing with the full benchmark when ranking model pairs that differ by fewer than 4 points of accuracy (bottom left, gray background). Once enough examples are selected to distinguish such model pairs (bottom right, gray background), random sampling is competitive. See §3 for details and Figure 4 for comparisons to another existing measure.

Overall, our meta-evaluation provides guidance for navigating the trade-off between evaluation efficiency and reliability. Very small micro-benchmarks have value, but it is vital to know that they will often only be able to distinguish models with very different performances. For the more pertinent task of distinguishing models with similar performances, larger micro-benchmarks are necessary, at which point random sampling is often enough for reliable, simple, and efficient evaluation.

## 2 MICRO-BENCHMARKING PRELIMINARIES

We give a formal description of micro-benchmarking using the terminology from Vivek et al. (2024). Given a large evaluation dataset $D_{\text{full}}$, the goal is to select a micro-benchmark $D_{\text{micro}} \subseteq D_{\text{full}}$ where $|D_{\text{micro}}| \ll |D_{\text{full}}|$. Micro-benchmark selection typically assumes access to a set of source models $\mathcal{U}$ that have been evaluated on the full dataset $D_{\text{full}}$. A method selects a micro-benchmark $D_{\text{micro}}$ of a fixed size $|D_{\text{micro}}| = n$ with a common high-level goal: for a new set of target models $\mathcal{T}$, performance on the micro-benchmark $D_{\text{micro}}$ should be similar to performance on the full benchmark. This goal is realized in various ways by different methods.

**Micro-benchmark selection.** We consider four micro-benchmark selection methods throughout this paper. Anchor Points (Vivek et al., 2024) first calculates correlations between example pairs using source model confidences and then selects the centroids of test example clusters in the resulting embedding space, in order to obtain a high correlation between model rankings on the full benchmark and the selected micro-benchmark. The tinyBenchmarks IRT method (Polo et al., 2024) instead aims to select a micro-benchmark that minimizes the error when predicting accuracy for individual source models. It does so by training an Item Response Theory (IRT) model, which results in example and source model embeddings. It then selects the closest example to centroids obtained from clustering these embeddings.[1] Stratified random sampling (Fogliato et al., 2024b) randomly selects examples from clusters obtained based on model confidence on examples. We also consider a diversity-based method that samples examples spread evenly in the space of source model correlations used by Anchor Points, enabled by a sampler that can select negatively-dependent samples (Bardenet et al., 2024). Appendix B gives detailed descriptions of these methods.

**Existing meta-evaluations for micro-benchmarks.** Prior work has measured the degree to which micro-benchmarks preserve target model performance in two ways: (i) for individual models using *mean estimation error* and (ii) in the aggregate for a whole set of target models using *rank correlation*. Mean estimation error measures the difference between model performance on the micro-benchmark and the full evaluation set $D_{\text{full}}$ on a set of target models $\mathcal{T}$ (Polo et al., 2024). On the other hand, Kendall's tau rank correlation (Nelsen, 2001) between all target model rankings on $D_{\text{micro}}$ and $D_{\text{full}}$ measures micro-benchmark fidelity on an entire set of target models (Vivek et al., 2024). A pair of models $M_1, M_2 \in \mathcal{T}$ is said to be a *discordant pair* if the models are ranked differently on the full benchmark and the micro-benchmark. Let $\mathsf{perf}_D(M)$ be the performance of model $M$ on an evaluation set $D$, and let $C$ be the set of these discordant pairs. The metrics are calculated as:

$$\operatorname*{err}_{D_{\text{micro}}, D_{\text{full}}} (\mathcal{T}) = \frac{1}{|\mathcal{T}|} \sum_{M \in \mathcal{T}} \Big| \operatorname*{perf}_{D_{\text{full}}}(M) - \operatorname*{perf}_{D_{\text{micro}}}(M) \Big| \qquad \text{Kendall's } \tau = 1 - \frac{2\,|C|}{\binom{|\mathcal{T}|}{2}} \qquad (1)$$

**Random sampling.** Uniform random sampling selects a fixed-size subset of examples independently and uniformly at random, without any model dependence:

$$D_{\text{micro}} \sim \text{Unif}\left( \left\{ R \subseteq D_{\text{full}} \big| |R| = n \right\} \right) \qquad (2)$$

We also consider another variant of stratified random sampling that takes into account a benchmark's $t$ pre-defined subtasks $\{D_i\}_{i=1}^{t}$ where $D_{\text{full}} = \bigcup_{i=1}^{t} D_i$, where an equal number of examples are selected from each subtask (Polo et al., 2024; Perlitz et al., 2024):

$$D_{\text{micro}} = \bigcup_{i=1}^{t} R_i \quad \text{where} \quad R_i \sim \text{Unif}\left( \left\{ R_i \subseteq D_i \big| |R_i| = \lfloor n/t \rfloor \right\} \right) \qquad (3)$$

**Micro-benchmark settings.** Many choices go into building micro-benchmarks: which source models to use, which target models to evaluate, and even which examples to select from in the first place. Prior work typically averages evaluation metrics over many partitions of a fixed set of models into source and target models at random (Vivek et al., 2024; Polo et al., 2024), by model family (Vivek et al., 2024), or by model release date (Polo et al., 2024). The examples selected for a micro-benchmark also vary over optimization hyperparameters, such as random seeds.

---

[1] Polo et al. (2024) propose method variants that also incorporate IRT model predictions. We focus on their core method, as our initial experiments showed similar results for variants.

# 3 MDAD: A META-EVALUATION FOR MICRO-BENCHMARK RELIABILITY

We present a meta-evaluation for micro-benchmarks based on how consistently pairwise model rankings on them agree with those obtained on the full benchmark. Given that model $M_1$ outperforms model $M_2$ on the full dataset, how likely is it that model $M_1$ also outperforms model $M_2$ on the micro-benchmark? We consider this pairwise ranking agreement probability as a function of the performance difference between the model pair, aggregated over all target model pairs.

We first calculate the probability of agreement between a micro-benchmark and the full dataset at a given difference in performance between two models on the full dataset. Let the performance difference between models $M_1$ and $M_2$ on eval dataset $D$ be $\Delta_D(M_1, M_2) = \mathsf{perf}_D(M_1) - \mathsf{perf}_D(M_2)$. We fix a set of ordered buckets of pairwise performance differences $\mathcal{B}$ and define agreement for each bucket $B \in \mathcal{B}$. Assuming without loss of generality that the model with higher performance on the full benchmark is called $M_1$, then for $D_{\mathrm{micro}}$ selected from $D_{\mathrm{full}}$:

$$\mathrm{agreement}\big(D_{\mathrm{micro}}, D_{\mathrm{full}}, B\big) = \Pr_{M_1, M_2 \in \mathcal{T}} \Big( \Delta_{D_{\mathrm{micro}}}(M_1, M_2) > 0 \,\Big|\, \Delta_{D_{\mathrm{full}}}(M_1, M_2) \in B \Big) \quad (4)$$

In practice, we compute agreement using all pairs of target model performances and calculate the frequency of target model comparisons in each bucket that match on both $D_{\mathrm{micro}}$ and $D_{\mathrm{full}}$.

The agreement function can be summarized using a single value that we call the *Minimum Detectable Ability Difference* (MDAD): what is the lowest performance difference on the full benchmark at which pairwise model rankings on a micro-benchmark are consistent with those on the full benchmark? Our goal here is to adapt the idea from statistical power analysis of estimating the minimum difference in model performance that can be consistently detected by a dataset (Card et al., 2020; Cohen, 1962). Following conventions from statistical power analysis, we consider a judgment consistent if it is correct at least 80% of the time:[2]

$$\mathrm{MDAD}(D_{\mathrm{micro}}, D_{\mathrm{full}}) = \underset{\substack{\mathrm{centroid}(B) \\ B \in \mathcal{B}}}{\arg\min} \Big\{ \mathrm{agreement}\big(D_{\mathrm{micro}}, D_{\mathrm{full}}, B\big) \Big\} \;\; \text{s.t.} \;\; \Pr \geq 0.8 \quad (5)$$

In practice, we report the centroid of the bucket corresponding to the lowest performance difference. MDAD captures how well a micro-benchmark preserves pairwise model rankings, unlike mean estimation error, which only considers single models, or rank correlation, which considers all model

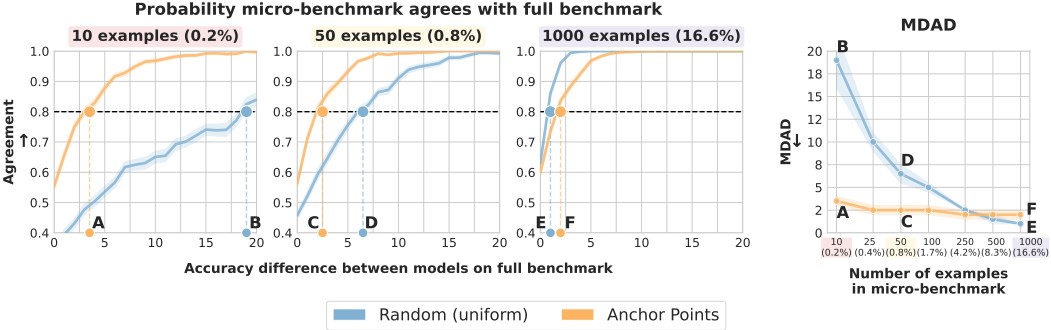

Figure 2: Agreement and MDAD measures on MMLU-Pro for uniform random sampling and Anchor Points with 300 source models. The three left panels show the probability that a pairwise ranking of models on a micro-benchmark agrees with the full benchmark's ranking, as a function of the accuracy difference between those models on the full benchmark. The rightmost panel summarizes all these agreement curves by showing the minimum detectable accuracy difference between models at each micro-benchmark size, i.e. the accuracy difference at which each curve in the first three panels crosses the 0.8 probability of agreement threshold. Points A-F show how each agreement curve is summarized by MDAD: each point marks the minimum difference in accuracy where an agreement curve surpasses a 0.8 probability of agreement. For MDAD, lower values are better. Error bars represent 95% bootstrap confidence intervals over 50 trials.

---

[2]Appendix C shows qualitatively similar results for different thresholds.

Table 1: Summary of differences between MDAD and existing meta-evaluation measures.

| Meta-evaluation measure | What does it measure? | Unit of comparison | Aggregation |
|---|---|---|---|
| Mean estimation error | Raw performance | Individual model | Across all models |
| Kendall's tau rank correlation | Model rankings | Model pair | Across all model pairs |
| **MDAD** | Model rankings | Model pair | Model pairs split by performance difference |

rankings in the aggregate. Lower MDAD is better because then even small performance differences across model pairs can be reliably measured under $D_{\text{micro}}$. If two models' performances differ by less than a micro-benchmark's MDAD, then that micro-benchmark is not likely to be able to consistently distinguish them. For example, suppose a micro-benchmark at a given size results in an MDAD of 10. Then at least 80% of the time, that micro-benchmark can correctly rank only model pairs that differ on the full benchmark by at least 10 performance points. If we instantiate agreement (Equation 4) using accuracy as the performance measure, Equation 5 then measures the *Minimum Detectable Accuracy Difference*. Figure 2 provides an illustrative example of the relationship between agreement at various accuracy differences and MDAD, which summarizes the agreement curve. For the rest of the paper, MDAD will refer to this accuracy-based instantiation. Table 1 gives a conceptual overview of how MDAD differs from existing meta-evaluation measures.

## 4 EXPERIMENTAL DESIGN

Our goal is to measure the reliability of micro-benchmarks as a function of which pairwise model rankings on a micro-benchmark predict pairwise model rankings on a) the full eval set and b) a fresh draw of equal size from the same task distribution.

For all experiments, we simulate different draws from a benchmark by splitting each benchmark in half (each subtask is divided in half as well): the train half is used to select the micro-benchmark, and the held-out half is used to measure generalization. We also uniformly at random partition a set of models into source models for selecting micro-benchmarks and a set of target models whose accuracy we are predicting, as in Vivek et al. (2024) and Polo et al. (2024). We account for variance in micro-benchmark construction by averaging over random samples of datasets and source models, following Card et al. (2020) and Perlitz et al. (2024). Most of our experiments select a micro-benchmark from an entire benchmark (as in Polo et al. (2024)), though in §5.4 we also evaluate selecting micro-benchmarks *per subtask* (as in Vivek et al. (2024)) and report averages across subtasks. We compute meta-evaluation measures with the same data split used for micro-benchmark selection for most experiments, following prior work. In §5.4, we do so using the held-out set. We use 50 trials, each with a partition of a) data points into a set for selection and a set for measuring generalization and b) models into source and target sets. Details are in Appendix D. Appendix E analyzes MDAD estimates for up to 100 trials; we find MDADs have stabilized by 50 trials.

**MDAD implementation details.** Following prior work, we consider micro-benchmarking methods specifically designed for classification tasks. We report accuracy as a percent from 0 to 100 and measure agreement using accuracy difference buckets at a resolution of 0.5 points of accuracy, i.e. $\mathcal{B} = \{[0, 0.25), [0.25, 0.75), [0.75, 1.25), \dots\}$, reporting MDAD as the bucket centroid.[3] In our experiments, MDAD takes on average 2.40 seconds to compute (with a standard deviation of 0.24 s).

**Benchmarks and micro-benchmarks.** We consider 47 subtasks of MMLU (10,631 examples, Hendrycks et al., 2021), all 24 subtasks of BIG-Bench Hard (BBH, 5,761 examples, Suzgun et al., 2023), all 14 subtasks of MMLU-Pro (12,032 examples, Wang et al., 2024), and GPQA (448 examples, Rein et al., 2024). We investigate four micro-benchmark selection methods discussed in §2: Anchor Points, tinyBenchmarks, stratified sampling by confidence, and diversity. We also compare to uniform random sampling and subtask-stratified random sampling.

**Size of micro-benchmarks.** We evaluate constructing micro-benchmarks at various small sizes by selecting $k \in \{10, 25, 50, 100, 250, 500, 1000\}$ examples. For GPQA, a smaller benchmark, we select $k \in \{10, 25, 50, 100, 200\}$ examples. In Appendix G, we also evaluate

---

[3]Appendix F shows that bucket resolutions of 0.25, 0.5, and 1 all yield similar MDAD values.

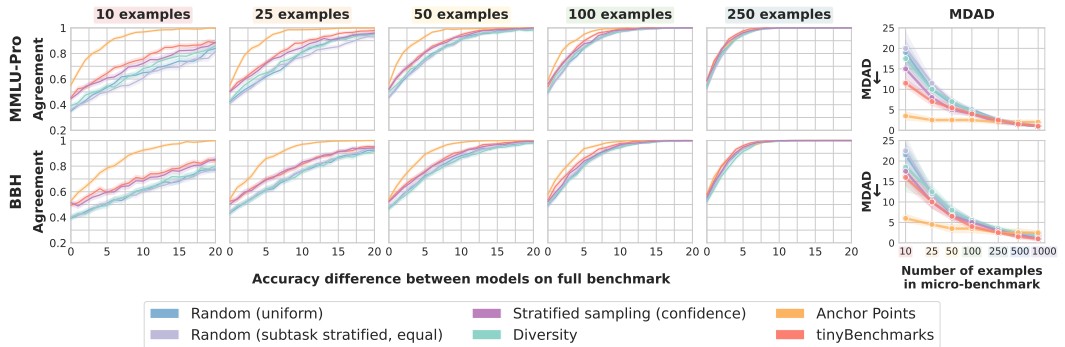

Figure 3: Comparing six micro-benchmarking approaches on two benchmarks. $y$-axis shows agreement (Equation 4), the probability that a micro-benchmark agrees with the full benchmark when comparing two models, as a function of how much those models differ on the full benchmark ($x$-axis). The rightmost column summarizes agreement curves using MDAD (Equation 5). For small micro-benchmarks, all methods struggle to compare models that differ by fewer than 4 points of accuracy on the full benchmark. Anchor Points does best, followed by tinyBenchmarks. Error bars show 95% bootstrap confidence intervals over 50 trials. Figure 9 (Appendix G) shows all benchmarks.

constructing micro-benchmarks at various proportions of the original benchmark by selecting $\{2\%, 4\%, 8\%, 16\%, 32\%, 40\%\}$ of examples, finding qualitatively similar results.

**Models.** For BBH, MMLU-Pro, and GPQA, we use the results of 470 models tagged as official on the Open LLM Leaderboard v2 (Fourrier et al., 2024), as evaluated with the LM Eval Harness (Gao et al., 2024). For MMLU, we use the results of the 366 models from the Open LLM Leaderboard (Fourrier et al., 2024), as in Polo et al. (2024). Model accuracy spans large ranges for all benchmarks, ranging from 25 to 75 on BBH, 27 to 76 on MMLU, 10 to 59 on MMLU-Pro, and 21 to 45 on GPQA. Unless otherwise stated, we randomly partition models into source models and target models as in (Vivek et al., 2024; Polo et al., 2024). We train micro-benchmarks with $\{10, 50, 100, 150, 200, 250, 300\}$ source models in order to determine whether increasing the number of source models substantially improves micro-benchmarks. Prior work has typically used a fixed number of source models for most experiments, either 10 source models (Vivek et al., 2024) or nearly 300 source models (Polo et al., 2024). Figures report results using 300 source models unless otherwise stated. Our approach of freshly computing micro-benchmarks using publicly available cached model predictions follows prior work (Polo et al., 2024).

**Difference from standard evaluations.** Our experiments do not attempt to find the "best" examples for evaluation—rather, they are designed to assess the reliability of existing micro-benchmarks. We also specifically seek to understand the conditions under which random sampling is an effective alternative to existing micro-benchmarking. For this reason, we do not release specific subsets of benchmarks like tinyMMLU (Polo et al., 2024) or Flash-HELM (Perlitz et al., 2024).

## 5 RESULTS

We consider four micro-benchmark selection methods and two random sampling baselines across the four benchmark suites, using agreement (Eq. 4) and MDAD (Eq. 5) from §3. MDAD reveals limitations of all the micro-benchmarking approaches we consider in the extreme dataset reduction regime and provides a finer-grained analysis than existing meta-evaluation measures (§5.1, §5.3). We show that random sampling is competitive with other micro-benchmark selection if sampling at least 250 examples (§5.2). §5.4 analyzes micro-benchmark generalization to new draws of the task. The complete results per benchmark for all parameter settings are in Appendix G. Appendix H shows that our analysis also holds for a micro-benchmarking method that selects whole subtasks.

**Larger micro-benchmarks afford lower MDAD.** Figure 3 examines how agreement varies with micro-benchmark size across methods and benchmarks. Each agreement curve is summarized by an

MDAD value (Figure 3, rightmost column); lower MDAD values correspond to higher reliability. As a first case study, consider BBH micro-benchmarks (Figure 3, bottom row). As more examples are selected, model pairwise rankings on micro-benchmarks are likely to be more predictive of those on the full eval set. For example, the agreement curves in the 100-example panel have shifted to the left of their positions in the 10-example panel. Just as agreement steadily increases as more examples are selected, MDAD steadily decreases for all methods as more examples are selected. As the number of examples increases from 10 to 100 to 1000, MDAD for tinyBenchmarks drops from 16 to 4 to 1.

**All evaluated micro-benchmarks have limits at extremely small sizes.**   The leftmost column of Figure 3 shows that BBH micro-benchmarks of size 10 cannot reliably rank model pairs unless they differ by almost 15 points of accuracy! The only exception to this is Anchor Points, which has higher agreement at smaller micro-benchmark sizes. When 10 examples are selected from BBH, Anchor Points achieves an MDAD of 6. If a model pair differs by more than 6 points on the full BBH, then this micro-benchmark is likely to correctly rank these models. If a model pair has an accuracy difference on the full BBH of less than the MDAD, e.g. 2 points of accuracy, then this micro-benchmark is unlikely to correctly rank these models. Overall all micro-benchmarking methods are limited when selecting only 10 examples. No method can consistently distinguish models that differ on the full benchmark by fewer than 3 points of accuracy on MMLU, 3.5 points of accuracy on MMLU-Pro, 6 points of accuracy on BBH, or 6.5 points of accuracy on GPQA.

**Anchor Points has the lowest MDAD at the smallest sizes but stagnates.**   The rightmost column of Figure 3 shows that Anchor Points has lower MDAD than other methods when selecting extremely few examples. We suspect this is because its method of selecting examples based on correlations in source model confidence is more closely aligned with what MDAD measures. On the other hand, Anchor Points has the highest MDADs when selecting 1,000 examples. We hypothesize this is due to very imbalanced cluster sizes in the underlying clustering that it performs. Anchor Points works by first clustering all of the full benchmark's examples using $k$-medoids and then selecting one example from each cluster. When selecting 10 examples from MMLU-Pro, cluster sizes are relatively even. But when selecting 1,000 examples, there is an extreme size imbalance between the 1,000 clusters of the benchmark examples: 47% of clusters are singletons, and the largest 10% of clusters together contain half of all the examples. Contrast this with tinyBenchmarks, which also performs clustering but uses $k$-means rather than $k$-medoids and uses a different embedding space. In the same setting of selecting 1,000 examples from MMLU with tinyBenchmarks, only 5% of clusters are singletons, and the largest 10% of clusters only contain 21% of examples. Each selected example from the large clusters in Anchor Points must stand in for many more data points.

## 5.1  MDAD AFFORDS FINER-GRAINED ANALYSIS THAN EXISTING MEASURES

Figure 4 compares MDAD to mean estimation error and Kendall's tau rank correlation using the experimental design from §4. MDAD provides a complementary—but not contradictory—perspective to these existing measures. We also report the correlation between MDAD and other meta-evaluation measures, using Kendall's tau rank correlation.[4] Across all settings, MDAD has a $\tau = 0.701$ ($p < 0.05$) Kendall's tau rank correlation with mean estimation error. MDAD also has a $\tau = -0.787$ ($p < 0.05$) correlation with the Kendall's tau rank correlation evaluation measure. Despite the high correlation and similar trends at a high level, e.g. Anchor Points stands out on both MDAD and rank correlation, there are still fine-grained differences between the measures. MDAD does not map one-to-one to other measures.

**MDAD provides more granular information than rank correlation.**   Even when micro-benchmarks have similar rank correlations in the aggregate, MDAD can identify fine-grained differences between methods. For example, tinyBenchmarks and uniform random sampling both have identical rank correlations when selecting 10 examples from MMLU-Pro (Points A and B, Fig. 4). But uniform random sampling has a much higher MDAD than tinyBenchmarks in this setting (C and

---

[4]Note here that we are using Kendall's tau rank correlation in a new way, to compare MDAD to existing meta-evaluation measures. One of these measures is itself the Kendall's tau rank correlation between micro-benchmark and full benchmark model rankings that we have been considering throughout. Each setting with a combination of benchmark, micro-benchmark size, and micro-benchmarking method yields a set of meta-evaluation measure values. We calculate the correlation between the rankings of settings induced by those values.

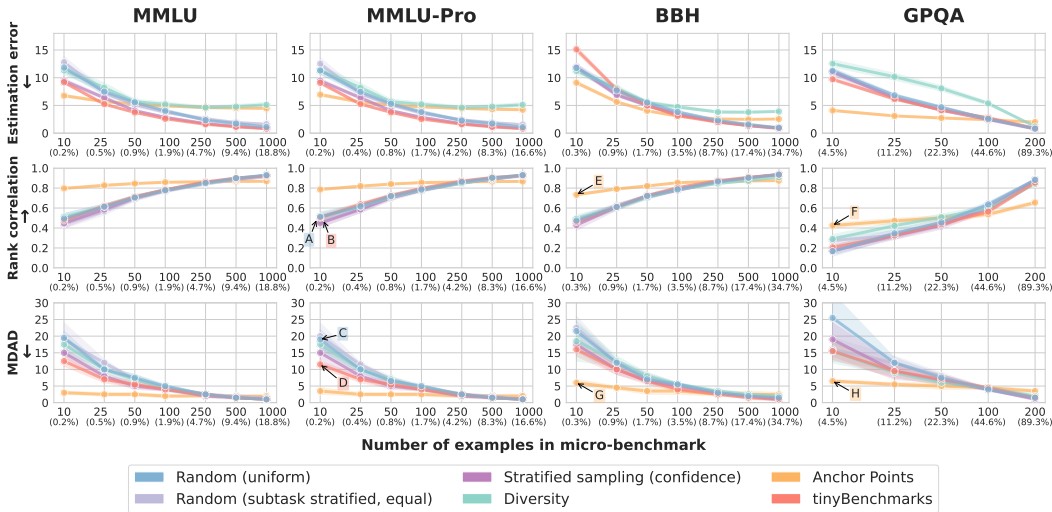

Figure 4: MDAD gives more granular information than mean estimation error and Kendall's tau rank correlation. Anchor Points is the only method that consistently outperforms random sampling at small dataset sizes across all metrics. Top row: Mean estimation error. Middle row: Kendall's tau rank correlation. Bottom row: Minimum Detectable Accuracy Difference (MDAD, ours, Equation 5). MDAD panels are the same as in Figure 3, shown here for ease of comparison. Points A-H labeled for ease of reference in §5.1. Error bars represent 95% bootstrap confidence intervals over 50 trials.

D, Fig. 4). Comparing across datasets shows that different rank correlations can map to the same MDAD value. When selecting 10 examples, Anchor Points has a much higher rank correlation of 0.73 on BBH but a rank correlation of 0.43 on GPQA (E and F, Fig. 4). In both cases, MDAD is 6 (G and H, Fig. 4). Even though the rank correlations are very different, both micro-benchmarks afford consistently accurate model comparisons when the models differ by at least 6 points of accuracy. Considering model comparisons only in the aggregate, as rank correlation does, obscures this finer-grained analysis. Additionally, MDAD values have a concrete interpretation—namely, the minimum model performance difference that an evaluation dataset can distinguish at least 80% of the time—while Kendall's tau values are harder to interpret on their own.

**MDAD accounts for consistent errors across models, unlike mean estimation error.** Mean estimation error is defined for a single model; it does not take into account whether a micro-benchmark consistently overestimates or consistently underestimates model accuracy for different models. For a simple illustrative example, if mean estimation error is 5 points, then the micro-benchmark's accuracy for a single model will be off by 5 points on average. If when comparing two models, the micro-benchmark overestimates both of their accuracies by 5 points, the micro-benchmark can still yield the correct pairwise model ranking. MDAD accounts for this by directly measuring whether pairs of models are correctly ranked. For example, when 100 examples are selected from MMLU-Pro, Anchor Points has a higher estimation error than random sampling but a lower MDAD. Mean estimation error does not directly capture when models will be ranked correctly, but MDAD does.

## 5.2 RANDOM SAMPLING IS COMPETITIVE WITH OTHER METHODS

Figure 4 shows that all methods improve according to all metrics as more examples are selected. For each benchmark, there is a micro-benchmark size at which both random sampling baselines become competitive with the other micro-benchmark selection. For MMLU, MMLU-Pro, and BBH, this occurs around 250 examples, and for GPQA (a much smaller dataset to begin with), this occurs at 200 examples. When selecting this many examples, all MDADs are 2 or less. But it is worth noting that when selecting fewer examples, MDADs for all methods are higher. Even when the other methods outperform random sampling at smaller dataset sizes, their MDAD values show that they do not always consistently rank models that differ by few points of accuracy. Consider, for example, selecting 10 examples from MMLU: tinyBenchmarks (MDAD of 12.5) outperforms random sampling

(MDAD of 20) by having a lower MDAD, but this means it cannot consistently distinguish models that differ by fewer than 12.5 points of accuracy on the full MMLU. When tinyBenchmarks achieves an MDAD of 2 by selecting 500 examples from MMLU, random sampling also has an MDAD of 2.

**Case study: MDAD better distinguishes micro-benchmarks from random at $\leq$100 examples.**
If we were to only use Kendall's tau rank correlation, we would have trouble differentiating most methods from random sampling when selecting between 10 and 100 examples. Consider performance on MMLU for $\leq$100 examples (Figure 4, left column). At these extreme dataset reductions, methods like tinyBenchmarks and stratified sampling by confidence have similar Kendall's tau rank correlations to random sampling, indicating that they all rank models equally well *in the aggregate*. But these methods have lower MDADs than random sampling, showing they can more reliably distinguish models that differ by fewer points of accuracy. This observation also holds for the other benchmarks.

### 5.3  MDAD CAN INTERPRET WHICH MODEL COMPARISONS WILL BE PRESERVED

We have so far established that at small sizes, most micro-benchmarks can distinguish only those models whose performance differ greatly on the full benchmark. This holds when comparing target models across various sizes and training regimes. But what about when comparing a specific set of models that we expect might have more similar performances? We consider a case study of 32 instruction-tuned 8B-parameter models on MMLU-Pro.

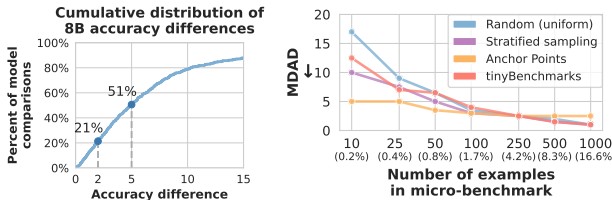

Figure 5: When comparing 8B-parameter instruction-tuned models on MMLU-Pro: model accuracies are in a narrow range, so nearly half of pairwise accuracy differences are less than 5 points (left), which is less than the MDAD for micro-benchmarks at small dataset sizes (right).

Most models have accuracies between 27 and 40 on the full benchmark, yielding very low pairwise accuracy differences (Figure 5, left). MDAD computed from these 8B-parameter models (Figure 5, right) helps us understand when model comparisons are preserved on micro-benchmarks of various sizes. All micro-benchmarks have an MDAD of 5 or more when selecting 10 or 25 examples. These micro-benchmarks are not likely to reproduce the full benchmark's ranking on the 51% of model comparisons that differ by at most 5 points of accuracy. When 1,000 examples are selected, most micro-benchmarks have an MDAD of 2. These can consistently rank more models, though they will still not be able to consistently rank the 21% of model pairs that differ by no more than 2 points of accuracy.

**MDAD explains why ranks stabilize when comparing models.**   Perlitz et al. (2024) observe that micro-benchmarks can often consistently predict the top-ranked target models, even when selecting few examples. MDAD offers a mechanism by which this occurs: top-ranked models often have high pairwise accuracy differences from many models, and micro-benchmarks often agree with the full benchmark when comparing very different models. That is, top-ranked models often differ from many models by more than a micro-benchmark's MDAD. While all micro-benchmarks have high agreement with the full benchmark for the top-performing models once 25 or more examples are selected, they are less likely to agree with the full benchmark when comparing the models in the middle of the distribution that are closer in accuracy to each other (Figure 14, Appendix L). Figure 5 (right) shows that when selecting 25 examples, all methods achieve an MDAD of 5 or more. For the top-ranked model, two-thirds of all model comparisons are above this MDAD. For a model in the middle of the distribution, only one third of all pairwise comparisons are above this MDAD. Appendix L shows similar results when restricting comparisons to various other sets of models, like instruction-tuned 70B-parameter models.

### 5.4  MICRO-BENCHMARKS GENERALIZE TO NEW EVALUATION SETS

One potential advantage of micro-benchmarks is that they may be able to exploit correlations between model predictions in order to predict how well models will do on fresh draws of the task. Or they could instead overfit their predictions to the specific examples they were selected from. We test

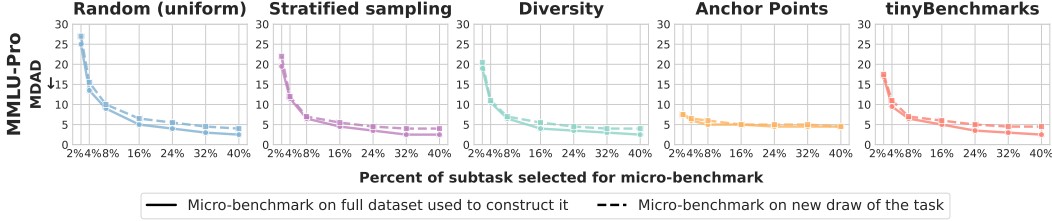

Figure 6: MDAD is modestly higher on MMLU-Pro when predicting relative model performance on a held-out draw of the task (dashed lines) than when predicting relative performance on the full dataset used to select the micro-benchmarks (solid lines). See Appendix J for results on other datasets.

whether either is the case by evaluating how well model comparisons on a micro-benchmark predict model comparisons on a held-out unseen set that was not used to construct the micro-benchmark. We find almost no difference in MDADs when predicting model performance on a held-out set using micro-benchmarks that select from entire benchmarks, indicating that these micro-benchmarking methods can effectively generalize to new draws of a dataset (full results in Appendix I). However, when selecting micro-benchmarks of subtasks individually, we do find that micro-benchmarks are slightly less able to predict model performance on fresh draws of the subtask, as evidenced by higher MDADs (Figure 6). Anchor Points experiences the least increase in MDAD when moving from the full benchmark to a fresh draw of the benchmark. Diversity and tinyBenchmarks experience larger increases. In all cases, MDAD is high. See Appendix J for full details and results.

## 6 DISCUSSION AND CONCLUSION

We have investigated how well various micro-benchmarking methods reproduce model performance judgments from full benchmarks using the meta-evaluation measures of agreement and Minimum Detectable Ability Difference (MDAD). We find that when micro-benchmarks produce model rankings with high aggregate correlation with the rankings from full benchmarks, they cannot always consistently distinguish model pairs with similar performance. Once enough examples are selected to allow for distinguishing model pairs with similar performance, random sampling is competitive with other methods. Our meta-evaluation measures can guide micro-benchmarking method designers in building reliable and efficient model comparisons. While our experiments primarily focus on accuracy-based evaluation, there exist straightforward extensions of MDAD to other metrics, including ones used in open-ended generation. In future work, we plan to explore using MDAD to directly guide data selection strategies.

We hope our meta-evaluation measures can help practitioners select the right micro-benchmark size for the job, rather than going with a one-size-fits-all recommendation from prior work. If the goal is to produce a ranking of models that may differ by, say, five points of accuracy or to track finer gradations in model performance over the course of training, evaluation sets should be large enough to afford a low MDAD. This is particularly relevant when seeking to identify if a model exceeds the current state-of-the-art, since each new model has historically achieved only incremental improvements in NLP (Card et al., 2020). However, if the goal is just to get a general sense of model performance, then even 10 examples chosen by Anchor Points or tinyBenchmarks from datasets like MMLU-Pro and BIG-bench Hard can suffice. Micro-benchmarks remain valuable tools for efficiently understanding model performance, but it is important to know the limits of that understanding.

## REPRODUCIBILITY STATEMENT

We release source code for reproducing all experiments, including all implementation details for datasets and micro-benchmarking methods, on GitHub. Appendix B describes the micro-benchmarking methods in detail, and Appendix D describes our experimental setup and infrastructure in greater detail. The GitHub repository also includes all intermediate micro-benchmarking results as well as the final results reported in figures throughout this paper.

ACKNOWLEDGMENTS

This research is supported in part by funding from the Allen Institute for AI, an Intel Rising Star Award and a USC Women in Science and Engineering (WiSE) Gabilan Fellowship. A portion of this work was done while S. Swayamdipta was a visitor at the Simons Institute for the Theory of Computing. We thank all members of the USC DILL Lab for constructive feedback, especially Matthew Finlayson and Sayan Ghosh. We also thank Kate Donahue, Varsha Kishore, Oliver Richardson, and Tejas Srinivasan for useful feedback.

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

## A    ADDITIONAL RELATED WORK

**Efficient evaluation.**    Benchmark datasets have been used to evaluate machine learning models for decades (Koch et al., 2021). Though benchmarks are not sufficient to fully map capabilities (Ethayarajh & Jurafsky, 2020; Kiela et al., 2021; Chiang et al., 2024), they remain an important tool for comparing models (Saxon et al., 2024). NLP benchmarks have grown ever-larger as LM capabilities have expanded (Srivastava et al., 2023; Liang et al., 2023). Micro-benchmarking methods select a small subset of a benchmark for evaluation with the goal of estimating a model's performance on the full benchmark. This is done by exploiting correlations in a model's predictions across examples (Fogliato et al., 2024b) and correlations across multiple models' predictions (Vivek et al., 2024; Ye et al., 2023; Liu et al., 2023), by training instance difficulty models using Item Response Theory (Polo et al., 2024; Vania et al., 2021; Rodriguez et al., 2021), or even by deduplication (Gupta et al., 2025). We study the reliability of micro-benchmarks that summarize performance across an entire dataset, though other work focuses on estimating performance across subtasks (Fogliato et al., 2024a). In contemporaneous work, Zhang et al. (2025) give complementary results on the efficacy of random sampling for predicting individual model accuracy. They find that a regression-based extension of random sampling has competitive mean estimation error with other micro-benchmarking methods when selecting 50 data points. They also study extrapolating from lower-accuracy source models to high-accuracy target models, finding that random sampling and a strategy based on augmented inverse propensity weighting achieve lower mean estimation error than other methods.

Farther afield, active testing instead selects test instances in an online per-model manner (Kossen et al., 2021). Still other works use model-based proxies to select unlabeled examples for annotation (Tahan et al., 2024; Zouhar et al., 2025). Subsets of *training data* can be selected using gradients (Everaert & Potts, 2024; Xia et al., 2024; Engstrom et al., 2025) or information gain (Deb et al., 2025), though such methods have not yet been applied to evaluation data selection.

**Evaluation reliability.**    For classification, larger evaluation datasets yield more reliable model comparisons (Shalev-Shwartz & Ben-David, 2014; Dror et al., 2018; Card et al., 2020) and are more robust to some forms of dataset reuse (Yauney & Mimno, 2024). Card et al. (2020) use statistical power to estimate the minimum detectable effect size afforded by a benchmark, though this approach requires assumptions about independence of examples, generalization error, and expected per-example agreement between model predictions. In contrast, our approach directly estimates the probability that pairwise model comparisons on a micro-benchmark agree with the full benchmark for a set of target models. Perlitz et al. (2024) also frame reliability as consistency over random choices in order to study the reliability of individual model error rate and model rankings over evaluation choices, but they do not identify which model comparisons are preserved by smaller datasets. Madaan et al. (2024) measure the variance in performance across random seeds and find that some micro-benchmarking methods increase variance. Our approach offers an interpretation of when variance impacts model comparisons. Other work considers the reliability afforded by LM generations for human evaluation (Ghosh et al., 2024; Boubdir et al., 2023).

## B    DETAILS OF MICRO-BENCHMARKING METHODS

**Anchor Points.**    We evaluate the "Anchor Points Weighted" method from Vivek et al. (2024). First, a correlation matrix $C$ between examples is constructed, where entry $C_{i,j}$ is a function of the correlation between examples $i$ and $j$ across all source models. $k$-medoids is used to select $n$ examples that maximize the correlation between the selected and the remaining examples. To estimate a target model's performance using these examples, the method averages the correct class probability of each example, weighted by the size of each example's corresponding cluster.

**tinyBenchmarks.**    We evaluate the "IRT" method from Polo et al. (2024), which trains an Item Response Theory model using `py-irt` (Lalor & Rodriguez, 2023) from source model predictions to produce embeddings for examples that are then clustered. After hyper-parameter sweeps, we use 10 dimensions for the IRT embeddings, and for IRT model training we use a learning rate of 0.1 and 2000 epochs.

**Stratified sampling by confidence.** We implement a variant of the stratified random sampling proposed by Fogliato et al. (2024b). We adapt the algorithm to work with multiple source models by taking the mean of model confidence across all source models. We perform $k$-means clustering into 10 strata based on these mean confidences. From each cluster, we uniformly at random sample a number of examples proportional to the size of that cluster. From model performance on this chosen subset of examples, we use the Horvitz-Thompson (HT) estimator (Horvitz & Thompson, 1952), as in Fogliato et al. (2024b), to arrive at an estimate of model performance on the full benchmark.

**Diversity.** We implement a method that selects diverse examples. The setup is similar to Anchor Points: each example has an embedding where each coordinate is a source model's confidence in the correct class. Rather than using $k$-medoids to select examples, we use the sampler from Bardenet et al. (2024) to select a diverse set of examples in this embedding space. Surprisingly, dimensionality reduction of embeddings to as few as 4 dimensions does not degrade performance.

## C    DIFFERENT AGREEMENT THRESHOLDS FOR MDAD

Figure 7 shows a comparison of 0.7, 0.8, 0.9, and 0.95 as the threshold of agreement in Equation 5. All of our other experiments use a threshold of 0.8. Agreement between a micro-benchmark and the full benchmark is higher for larger differences in model performance As the agreement threshold for MDAD increases, the main effect is that MDAD also increases. All methods tend to experience this MDAD increase, so the overall results for different thresholds are qualitatively similar.

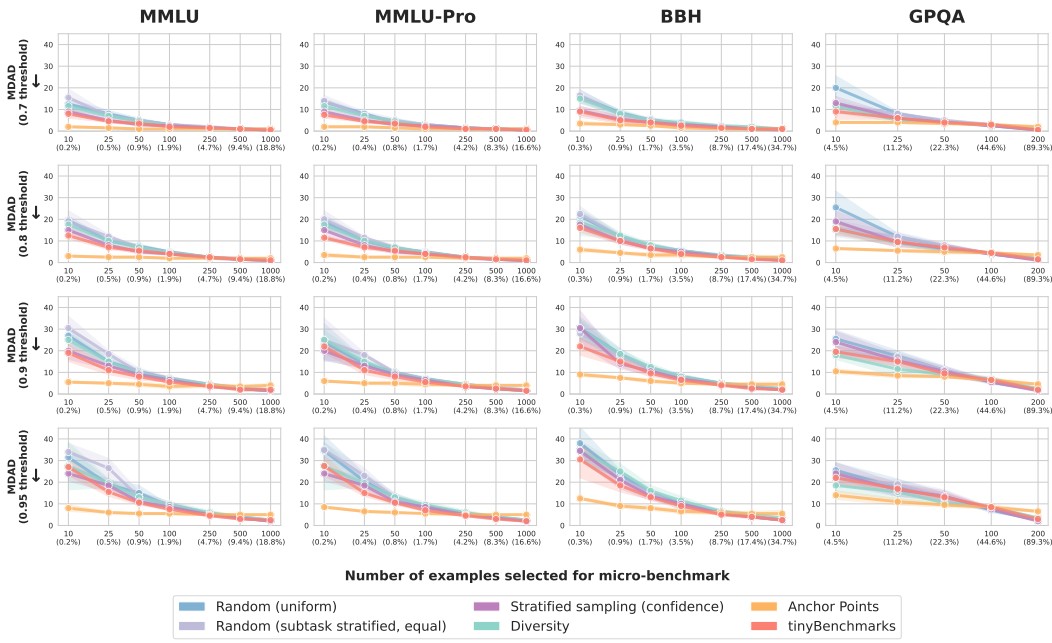

Figure 7: MDADs when using different thresholds for agreement are qualitatively similar. The second row of MDAD panels with a 0.8 threshold are the same as in Figure 4. Error bars represent 95% bootstrap confidence intervals over 50 trials.

## D    FULL EXPERIMENTAL SETUP AND COMPUTING INFRASTRUCTURE

For most experiments, we construct micro-benchmarks using 300 source models and evaluate them using 50 target models. We average all evaluation metrics over 50 runs of random partitions into source and target models. For the experiments in Section 5.2 and Appendix L, we choose 300 random source models and evaluate with a fixed set of (non-overlapping) target models. Experiments were implemented using NumPy (Harris et al., 2020). We report calculated MDADs to the nearest 0.5.

**Models.**    For MMLU, we use the results of the 366 models from the Open LLM Leaderboard (Fourrier et al., 2024), as in Polo et al. (2024). For the other benchmarks, we use cached model predictions from the 470 models tagged as official on the Open LLM Leaderboard v2 (Fourrier et al., 2024). For each of these benchmarks, we include models that had full per-example evaluations on all subtasks, yielding 447 models for MMLU-Pro, 409 models for BBH, and 420 models for GPQA. Our evaluations include 101 models with 0.5-3B parameters and 39 models with 70B+ parameters. The largest model included in our evaluations is 141B.

**Computing infrastructure.**    Micro-benchmarking methods and analysis were run on a cluster node with 4 GeForce GTX 1080 Ti GPUs and a Macbook Pro with an Apple M3 Pro processor and 18GB of RAM.

**Runtime.**    Table 2 gives the average time (in seconds) for each method on each benchmark to perform one trial with 300 source models. Our full experiments are 50 trials across 7 different settings for number of source models, for a total of 91.05 hours.

| Micro-benchmarking method | MMLU | MMLU-Pro | BBH | GPQA |
|---|---|---|---|---|
| Random (uniform) | 3.1 | 3.5 | 3.8 | 0.8 |
| Random (subtask stratified, equal) | 3.1 | 3.5 | 3.5 | 0.7 |
| Stratified sampling (confidence) | 3.0 | 2.8 | 2.9 | 0.5 |
| Anchor Points | 6.2 | 7.7 | 4.7 | 1.1 |
| tinyBenchmarks | 150.9 | 164.5 | 58.9 | 7.5 |
| Diversity | 315.0 | 266.2 | 207.0 | 16.8 |

Table 2: Average time (seconds) for completion of one trial.

## E    QUANTITATIVE ERROR ANALYSIS OF MDAD

Our results throughout the paper estimate MDAD across 50 trials of our meta-evaluation experiments. Table 3 shows estimated MDADs and 95% confidence intervals for up to 100 trials for uniform random sampling, Anchor Points, and tinyBenchmarks when selecting 50 and 100 examples from MMLU-Pro. Estimated MDADs have stabilized by 50 trials.

Table 3: MDADs with 95% confidence intervals for up to 100 trials for uniform random sampling, Anchor Points, and tinyBenchmarks when selecting 50 and 100 examples from MMLU-Pro.

| | 50 examples | | | 100 examples | | |
|---|---|---|---|---|---|---|
| Number of trials | Random (uniform) | Anchor Points | tinyBenchmarks | Random (uniform) | Anchor Points | tinyBenchmarks |
| 10 | $5.6 \pm 3.4$ | $2.90 \pm 2.1$ | $3.6 \pm 3.1$ | $4.6 \pm 2.7$ | $2.8 \pm 2.3$ | $3.6 \pm 1.7$ |
| 20 | $6.1 \pm 2.1$ | $3.00 \pm 1.7$ | $5.3 \pm 2.0$ | $4.5 \pm 1.5$ | $2.8 \pm 1.7$ | $3.6 \pm 1.5$ |
| 30 | $6.1 \pm 1.9$ | $3.50 \pm 1.2$ | $5.2 \pm 1.7$ | $4.5 \pm 1.6$ | $3.0 \pm 1.6$ | $3.6 \pm 1.4$ |
| 40 | $6.2 \pm 1.3$ | $3.50 \pm 1.2$ | $5.4 \pm 1.2$ | $4.4 \pm 1.3$ | $3.4 \pm 1.1$ | $4.0 \pm 1.3$ |
| 50 | $6.3 \pm 1.3$ | $4.10 \pm 1.3$ | $5.4 \pm 1.8$ | $4.4 \pm 1.3$ | $3.6 \pm 1.1$ | $3.8 \pm 0.8$ |
| 60 | $6.5 \pm 1.0$ | $4.10 \pm 1.3$ | $5.4 \pm 1.2$ | $4.4 \pm 1.0$ | $3.7 \pm 1.1$ | $3.8 \pm 0.8$ |
| 70 | $6.6 \pm 1.0$ | $4.00 \pm 1.1$ | $5.4 \pm 1.2$ | $4.4 \pm 0.6$ | $3.7 \pm 1.1$ | $3.9 \pm 0.7$ |
| 80 | $6.6 \pm 1.0$ | $4.00 \pm 1.1$ | $5.3 \pm 1.0$ | $4.4 \pm 0.7$ | $3.6 \pm 0.9$ | $3.8 \pm 0.7$ |
| 90 | $6.6 \pm 0.9$ | $4.10 \pm 1.2$ | $5.2 \pm 0.8$ | $4.4 \pm 0.6$ | $3.7 \pm 0.9$ | $3.8 \pm 0.7$ |
| 100 | $6.6 \pm 0.9$ | $4.10 \pm 1.2$ | $5.2 \pm 0.8$ | $4.4 \pm 0.6$ | $3.7 \pm 0.9$ | $3.7 \pm 0.6$ |

# F    DIFFERENT BUCKET RESOLUTIONS FOR MDAD

Figure 8 shows a comparison of 0.25, 0.5, and 1.0 as bucket resolutions when calculating MDAD (Equation 5). All of our other experiments use a resolution of 0.5. These different resolutions result in very similar MDADs in most cases. The one exception is when selecting 10 examples from GPQA with random sampling: agreement values in the larger buckets are so low that no bucket has an agreement value greater than 0.8.

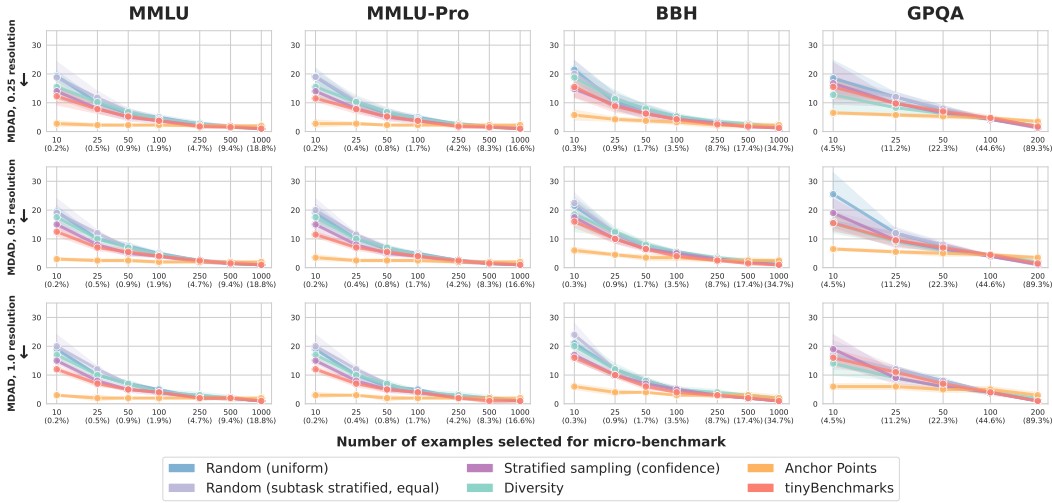

Figure 8: MDADs when using different bucket resolutions for agreement are qualitatively similar. The middle row of MDAD panels are the same as in Figure 4. Error bars represent 95% bootstrap confidence intervals over 50 trials.

# G   FULL RESULTS

Figure 9 is an expanded version of Figure 3 with full correctness curves for all datasets. Figure 10 gives results when selecting a fixed percentage of examples from each subtask in a benchmark.

Figure 16 (at the end of the appendix) gives full results for all benchmarks when selecting a fixed number of examples from each benchmark. Figure 17 (also at the end of the appendix) gives full results for all benchmarks when selecting a fixed percentage of examples from each benchmark.

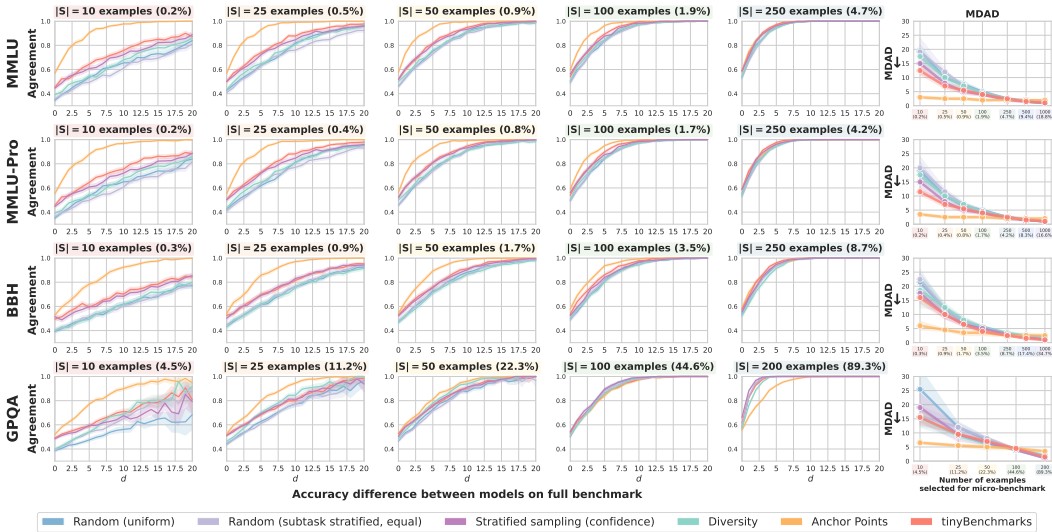

Figure 9: Expanded version of Figure 3 with all datasets. Comparing six micro-benchmarking approaches on four evaluation benchmarks. $y$-axis reports agreement, the probability that a micro-benchmark agrees with the full benchmark when comparing two models, as a function of how much those models differ on the full benchmark ($x$-axis). The rightmost column summarizes agreement curves using MDAD. Error bars represent 95% bootstrap confidence intervals over 50 trials.

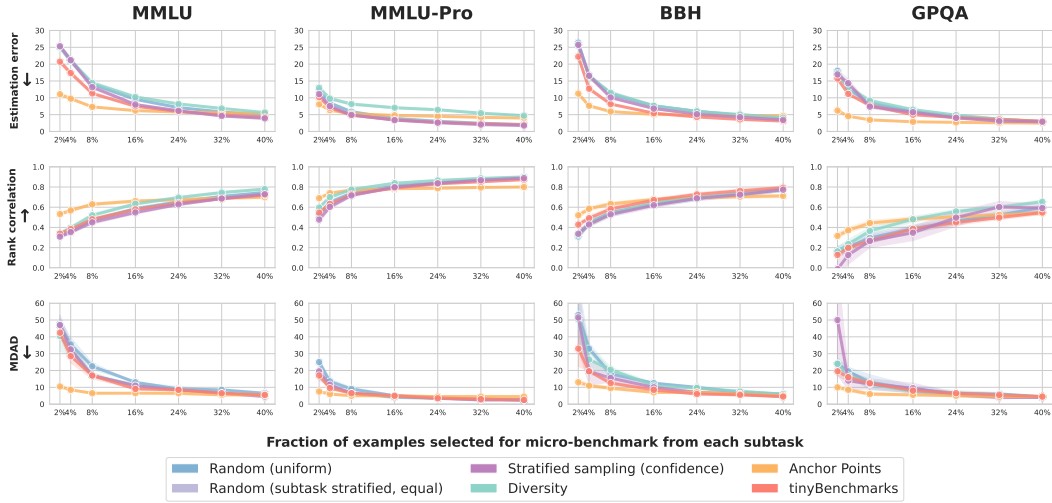

Figure 10: Results when selecting a fixed percentage of examples from each subtask of a benchmark. Top row: Mean estimation error. Middle row: Kendall's tau rank correlation. Bottom row: Minimum Detectable Accuracy Difference (MDAD, ours). Results are averaged over all subtasks. Error bars represent 95% bootstrap confidence intervals over 10 trials.

## H MDAD CAN ANALYZE METHODS THAT SELECT ENTIRE SUBTASKS

Throughout this paper, we evaluate micro-benchmarking methods that select examples without regard to which subtasks of the original benchmark they are from. But our methods and proposed meta-evaluation measures generalize to other kinds of micro-benchmarking methods as well. For example, BenTo is a micro-benchmarking method that selects whole subtasks for smaller evaluation sets by estimating task transferability (Zhao et al., 2025). When selecting a micro-benchmark from MMLU, BenTo selects 801 examples in three subtasks of MMLU for micro-benchmarking. We calculate MDAD and other meta-evaluation measures for this selected subset of examples, as well as uniform random sampling for selecting 801 examples (Table 4). Both BenTo and Random achieve very similar results.

Table 4: Results for the BenTo micro-benchmarking method.

| Method | Mean estimation error | Kendall's tau rank correlation | MDAD |
|---|---|---|---|
| BenTo | $1.34 \pm 0.39$ | $0.9332 \pm 0.0153$ | $1.5 \pm 0.5$ |
| Random (uniform) | $1.30 \pm 0.38$ | $0.9166 \pm 0.0175$ | $1.5 \pm 0.5$ |

## I GENERALIZING TO NEW TASK DRAWS WHEN SELECTING FROM ENTIRE BENCHMARKS

Figure 11 gives results for meta-evaluating micro-benchmarks with respect to a held-out set when selecting from the benchmark as a whole. MDADs can increase by up to 0.5 points in this setting.

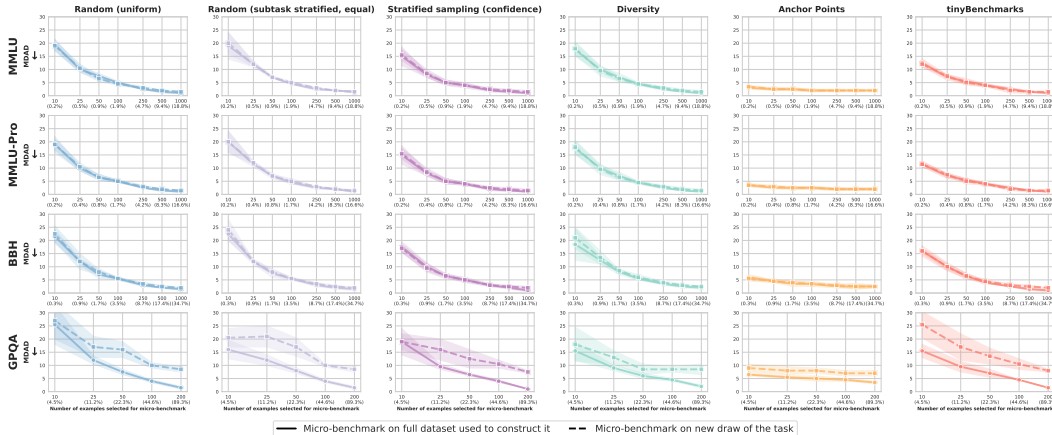

Figure 11: For all methods on MMLU, MMLU-Pro, and BBH, MDAD is nearly the same when predicting relative model performance on a held-out draw of a task (dotted lines) as when predicting relative performance on the full set of examples used to select the micro-benchmarks (solid lines). For GPQA, a much smaller dataset, there is a larger gap in performance. Error bars represent 95% bootstrap confidence intervals over 50 trials.

# J  GENERALIZING TO NEW TASK DRAWS WHEN SELECTING FROM SUBTASKS SEPARATELY

Figure 12 is an expanded version of Figure 6 with results for comparing micro-benchmarking methods on a held-out set when selecting from subtasks individually.

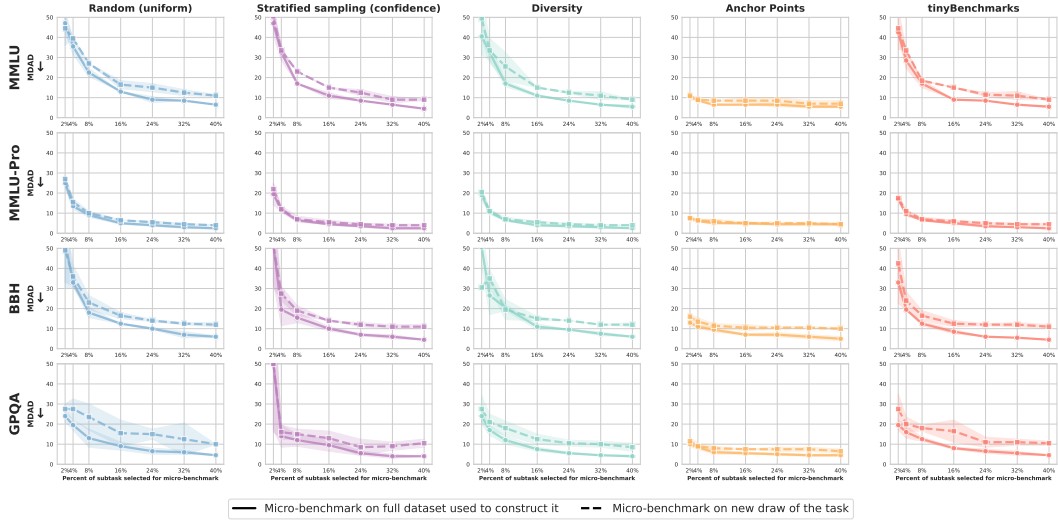

Figure 12: For all methods, MDAD is modestly higher when predicting relative model performance on a held-out draw of a task (dotted lines) than when predicting relative performance on the full set of examples used to select the micro-benchmarks (solid lines). The MMLU-Pro row is the same as in Figure 6. Error bars represent 95% bootstrap confidence intervals over 10 trials.

Table 5 gives changes in other meta-evaluation measures in this setting for MMLU-Pro. When moving from the original draw of the task to a new draw of the task, mean estimation error increases by less than 1 point, and Kendall's tau rank correlation decreases by less than 0.05 for all evaluated micro-benchmarking methods.

Table 5: Changes in mean estimation error and Kendall's tau rank correlation for MMLU-Pro when generalizing to new draws of the task, as averaged across all selected micro-benchmark sizes (corresponding to the MDADs in Figure 6, which are included here for reference).

| Method | Mean increase in MDAD | Mean increase in mean estimation error | Mean decrease in Kendall's tau rank correlation |
|---|---|---|---|
| Random (uniform) | 1.18 | 0.65 | 0.039 |
| Stratified sampling (confidence) | 1.12 | 0.75 | 0.039 |
| Diversity | 1.12 | 0.41 | 0.041 |
| Anchor Points | 0.37 | 0.31 | 0.014 |
| tinyBenchmarks | 1.07 | 0.74 | 0.038 |

We also find that comparisons between models at the same scale are not always preserved, but micro-benchmarks can still consistently distinguish the performance differences between smaller (7B) models and larger (70B) models. In Section 5.4, we find that the MDAD of micro-benchmarking methods can increase on a new draw of the dataset by up to 1.2 points when selecting examples from individual subtasks (Figure 6). When selecting 100 examples from MMLU-Pro, all micro-benchmarking methods have an MDAD of at most 5 when generalizing to a new draw of the task. When comparing models in the 6B-8B range to each other, 33.3% of comparisons will not be preserved by the micro-benchmarks because they involve accuracy differences below the MDAD. When comparing models in the 68B-72B range to each other, 35.7% of comparisons will not be preserved. But the micro-benchmarks can still consistently distinguish between small and large models because only 6.9% of those comparisons have accuracy differences less than 5.

# K    INCREASING NUMBER OF SOURCE MODELS HAS MODEST EFFECT

So far we have been evaluating how MDAD decreases as the number of selected examples increases. For micro-benchmarking method designers, a key parameter is how many source models to use when selecting the micro-benchmark. Whereas previously we have examined performance with 300 source models, Figure 13 shows aggregate results for all benchmarks for many different numbers of source models used to select the micro-benchmark. For nearly all datasets and numbers of examples selected, increasing the number of source models provides only modest improvement in model distinguishability for any of the methods. The effect of more source models is most pronounced for Anchor Points on BBH and GPQA when moving from 10 to 50 source models, though more source models do not yield further improvements. For all methods, increasing the number of source models is not as effective as evaluating on more examples.

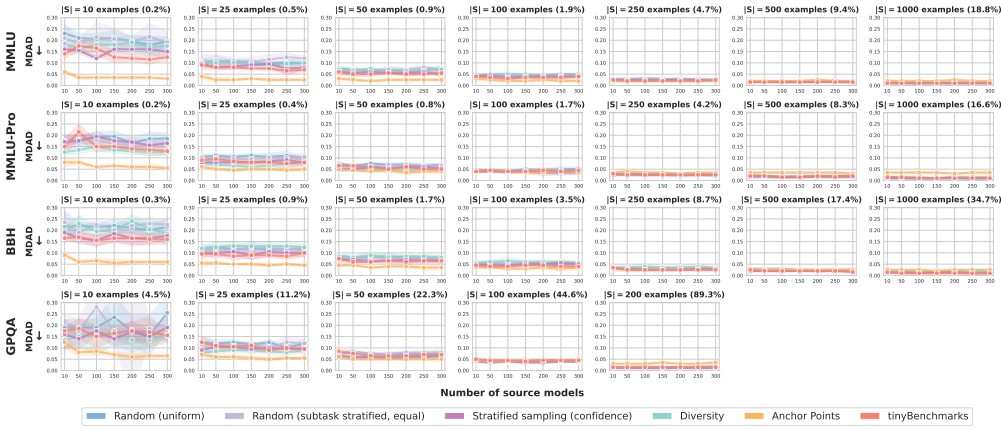

Figure 13: Allowing micro-benchmarking methods access to increasing numbers of source models with full predictions does not improve MDAD as much as evaluating on even slightly more examples, as indicated by horizontal lines in nearly all panels. Random sampling is provided as a baseline, as it does not rely on any source models. Note that the $x$-axis is number of source models, not number of examples selected. Error bars represent 95% bootstrap confidence intervals over 50 trials.

# L  COMPARING SPECIFIC SIZES OF MODELS

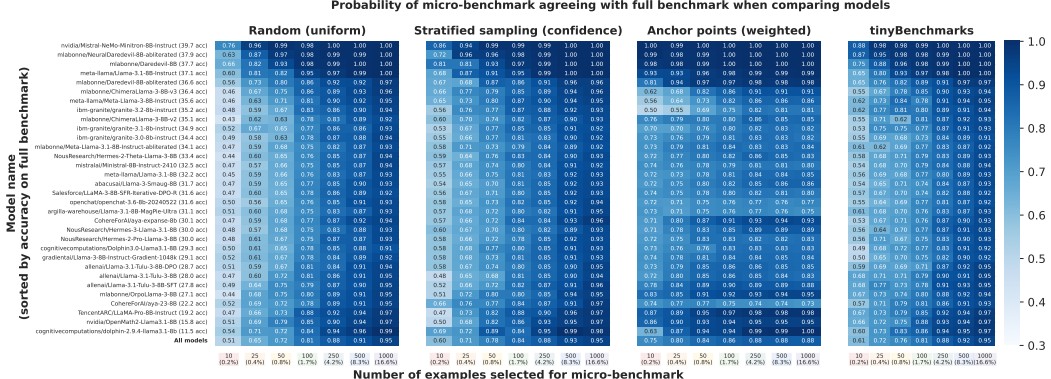

Figure 14: When comparing 8B-parameter instruction-tuned models on MMLU-Pro: per-model agreement with the full benchmark is lower for the models in the middle of the accuracy distribution that have more similar accuracies to many models.

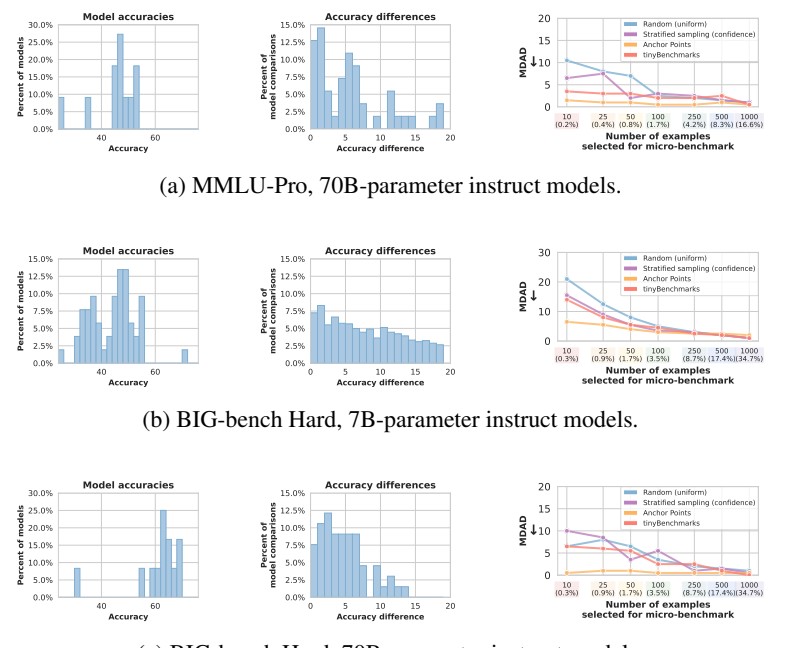

(a) MMLU-Pro, 70B-parameter instruct models.

(b) BIG-bench Hard, 7B-parameter instruct models.

(c) BIG-bench Hard, 70B-parameter instruct models.

Figure 15: Pairwise model comparisons are often between models with similar accuracies when comparing specific classes of models.

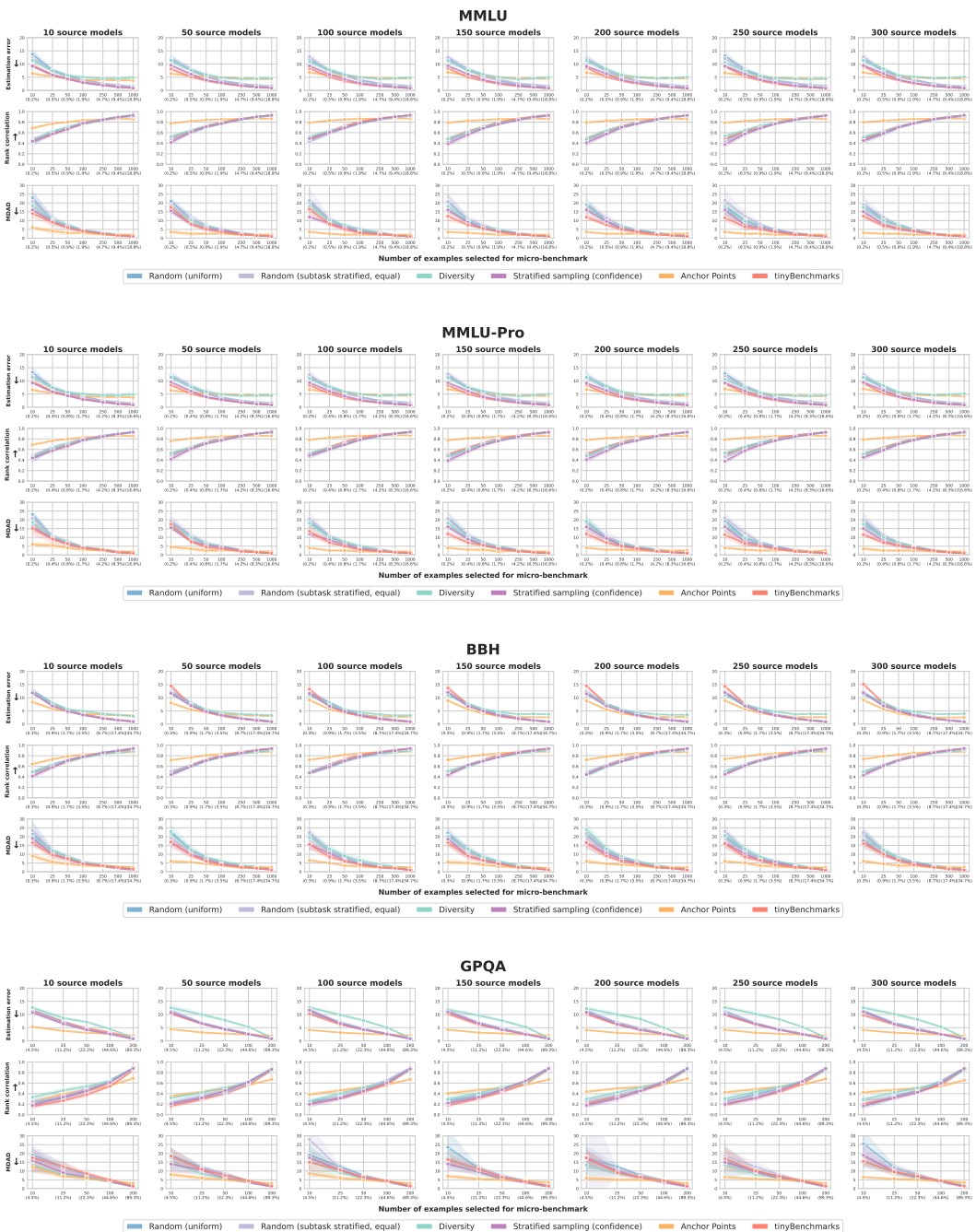

Figure 16: Full results when selecting a fixed number of examples from across entire benchmarks. The column with 300 source models is the same as the results presented in Figure 4. Top row: Mean estimation error. Middle row: Kendall's tau rank correlation. Bottom row: Minimum Detectable Accuracy Difference (MDAD, ours). Error bars represent 95% bootstrap confidence intervals over 50 trials.

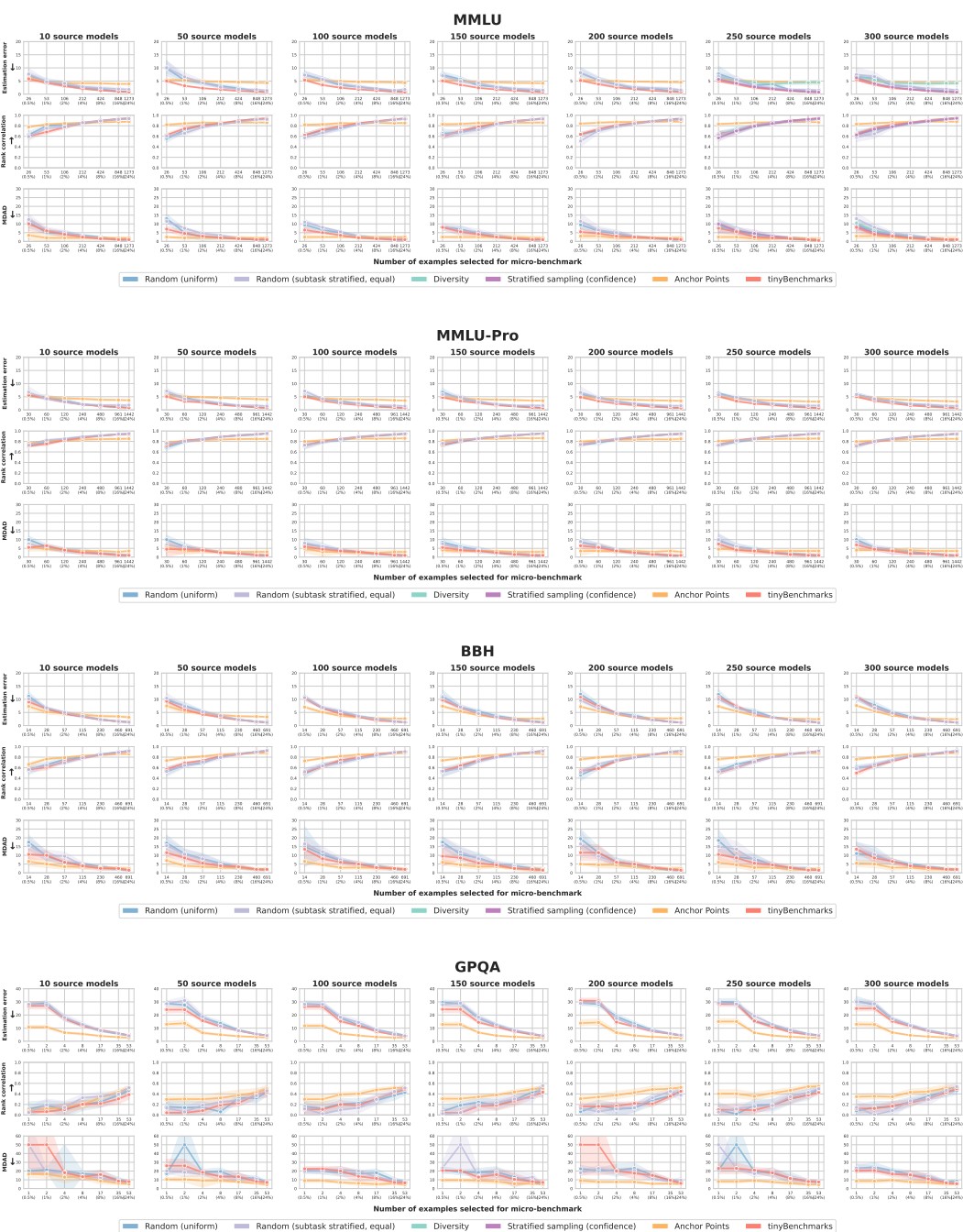

Figure 17: Full results when selecting a fixed *percentage* of examples from across entire benchmarks. Top row: Mean estimation error. Middle row: Kendall's tau rank correlation. Bottom row: Minimum Detectable Accuracy Difference (MDAD, ours). Error bars represent 95% bootstrap confidence intervals over 50 trials.

