# OpenReview forum: "How Reliable is Language Model Micro-Benchmarking?"
_ICLR.cc/2026/Conference — ICLR 2026 Oral_

### Official Review · Reviewer_E4xW · 2025-10-30

**Soundness:** 3
**Presentation:** 2
**Contribution:** 2
**Rating:** 6
**Confidence:** 3

**Summary:**

This work investigates the reliability of micro-benchmarks, small subsets of larger evaluation datasets, in faithfully ranking model performance. The authors introduce a new meta-evaluation metric, MDAD (Model Disagreement Agreement Distance), which quantifies how consistently a micro-benchmark preserves the pairwise ranking of models relative to the full benchmark. Unlike other meta-benchmarking strategies, the authors claim that MDAD provides a finer-grained view of evaluation stability, revealing when and why model rankings diverge under extreme data reduction. By analyzing MDAD across various subset selection strategies, the study highlights the trade-offs between sampling efficiency and ranking fidelity, offering practical insights into how different sampling techniques perform as dataset size decreases.

**Strengths:**

* The experiments demonstrate MDAD’s potential to offer deeper insights into selecting the most reliable micro-benchmarking approach (Fig. 3).
* The experiments further show that MDAD provides more fine-grained and informative analysis than existing meta-evaluation metrics for micro-benchmarks (Fig. 4).
* The authors appropriately situate their work within the broader literature, clearly articulating how their contributions extend prior research.

**Weaknesses:**

* Some figure titles and labels are difficult to read; improving their legibility would make the results clearer and more accessible.
* Given that meta-evaluations are conceptually dense and rely heavily on prior work, the paper would benefit from a thorough proofreading and clarity-focused revision to improve readability.
* Including comparative tables or summary diagrams that explicitly contrast this meta-evaluation method with existing approaches would help highlight its unique advantages.
* The paper’s contribution is summative and complementary, it extends and provides additional insights useful to decide on existing micro-benchmarking techniques rather than proposing an entirely new framework.
* The authors provide source code in the supplementary materials, but it remains unclear whether these materials will be publicly released to support transparency and reproducibility.

**Questions:**

* What potential applications beyond evaluation do you foresee for the proposed metric? Could MDAD, or a variant of it, be extended to support data selection strategies for supervised fine-tuning (SFT) or other training workflows?
* Which experimental assets will be publicly released, for example, code, sampled subsets, or analysis scripts? Please provide specific details regarding the planned scope and accessibility of these materials.

---

> ### Author Response · Authors · 2025-11-21
> **Author response to Reviewer E4xW**
>
> Thanks for your review!
>
> ***Weakness 1:** Figure text is hard to read.*
>
> We have increased the font sizes in all figures in the main paper.
>
> ***Weakness 2:** The paper would benefit from a thorough proofreading and clarity-focused revision to improve readability.*
>
> We have better delineated case studies in Section 5 at the suggestion of Reviewer h3U2, improved figure readability, and added the table summarizing MDAD’s differences from existing measures that you suggest in your Weakness 3 to reduce reliance on knowledge of prior work. We have also shortened some sentences for clarity throughout the paper and added three paragraph headers in Section 5 for improved organization. We hope these changes address your concerns about readability.
>
> ***Weakness 3:** Including comparative tables  that explicitly contrast this meta-evaluation method with existing approaches would help highlight its unique advantages.*
>
> Thanks for this suggestion. We have added Table 1 to Section 3 to summarize the differences between MDAD and existing meta-evaluation measures.
>
> ***Weakness 4:** The paper provides additional insights useful to decide on existing micro-benchmarking techniques rather than proposing an entirely new framework.*
>
> The goal of our paper is to provide a new tool that gives a more fine-grained analysis of micro-benchmarking. We argue throughout the paper that this tool is novel and allows for novel insights about micro-benchmarking techniques. We believe robust evaluation is vital for determining whether new micro-benchmarking techniques improve over existing methods, and we hope MDAD can also serve to inform data selection--see our response to your Question 1 below for possibilities.
>
> ***Weakness 5 / Question 2:** It remains unclear whether the supplementary materials will be publicly released. Which experimental assets will be publicly released, for example, code, sampled subsets, or analysis scripts?*
>
> All code, raw micro-benchmarking results, meta-evaluations, and analysis/figure-making scripts are included in the supplementary material and will be made publicly available on Github upon this paper’s publication. We will also make all the cached raw model predictions on all benchmarks available on Github. This Github repo will be linked in the abstract of the final paper. We have updated the reproducibility statement to explicitly state this.
>
> ***Question 1:** What potential applications beyond evaluation do you foresee for the proposed metric? Could MDAD, or a variant of it, be extended to support data selection strategies for SFT or other training workflows?*
>
> This is a great question. We agree that training data selection is a very interesting direction for future work; indeed we are actively working on data selection already. As a first step, we have some preliminary experiments that MDAD can inform micro-benchmark creation.
>
> We have implemented a variance-based greedy selection strategy that iteratively constructs low-MDAD micro-benchmarks. At each iteration, we score each remaining example by the variance of its pairwise performance differences across all model pairs. For instance, an example where all model pairs show similar performance (low variance) provides little discriminative information, while an example where some pairs differ greatly and others don't (high variance) is highly informative for ranking. In preliminary experiments, we compute MDAD for source model comparisons using the top scoring candidates, to select the example yielding the lowest MDAD when added to the running subset of examples (we also report another measure for a fairer comparison). This process continues until reaching the desired micro-benchmark size. When constructing a 50-example micro-benchmark from MMLU-Pro, this method has promising results, as shown below. Compared to other methods, our new method achieves lower MDAD (calculated with target models not used for selection) and lower mean estimation error. We are actively scaling up this method and its evaluation. This shows that MDAD can support data selection strategies in addition to being a meta-evaluation metric.
>
> | Method             |        MDAD  (&downarrow;)     |     Mean estimation error (&downarrow;)    |
> |-----------------------------------------|:-----------:|:-----------:|
> | **Variance-based Greedy Selection**  |    **2.50**   |  **3.06**    |
> | Random (uniform）                                |        7.50      |      5.95        |
> | Anchor Points                                          |        6.50      |      5.00        |
> | TinyBenchmarks IRT                               |        6.50      |      5.29        |

---

> > ### Comment · Reviewer_E4xW · 2025-11-22
> >
> > > W1 and W2
> >
> > Understood on the plan, but please add the updated paper version with updates tracked by color.
> >
> > > W3
> >
> > Thanks.
> >
> > > W4
> >
> > While summative I agree the contribution is useful.
> >
> > > W5
> >
> > ok.
> >
> > > Q2
> >
> > Thanks for sharing these insights, you could add a `future work` section to the paper centered on this point.

---

> > > ### Author Response · Authors · 2025-11-25
> > >
> > > Thanks for your response. Updates are now marked in blue in the latest pdf. For clarity, we don't color the larger text in Figures 1-6, though they have all been updated. We agree with your suggestion about future work and will consider adding this discussion of data selection to the final version of the paper.

---

### Official Review · Reviewer_1D3U · 2025-10-31

**Soundness:** 4
**Presentation:** 4
**Contribution:** 3
**Rating:** 8
**Confidence:** 5

**Summary:**

The paper presents Minimum Detectable Ability Difference (MDAD), a meta-evaluation measure. It uses MDAD to assess different micro-benchmarking methods, compare MDAD to other meta-evaluation measures, and analyze micro-benchmarking efficiency-reliability tradeoff.

**Strengths:**

1. The paper is well written and structured.
2. The proposed measure is well motivated and intuitive.
3. The experiments are well designed, including a few benchmarks, a large number of models, across different subset sizes.
4. The figures are effective in showcasing the relevant results.
5. Sections 5.1 and 5.3 are doing a good job and demonstrating why MDAD adds value above other methods.

**Weaknesses:**

1. Figures are hard to read.
2. The conclusions themselves are relatively known and expected. This is reasonable. The new measure and quantifiable results are new and interesting.

**Questions:**

The results echo a broader message of "the bitter lesson". Specified methods work well in small-scale settings (model sizes, number of examples, etc.) but do not generalize to larger scales. What is your perspective from that context? What is its relation to the micro-benchmarking methods?

---

> ### Author Response · Authors · 2025-11-21
> **Author response to Reviewer 1D3U**
>
> Thanks for your review!
>
> ***Weakness:** The conclusions themselves are relatively known and expected. This is reasonable. The new measure and quantifiable results are new and interesting.*
>
> We appreciate that you find the measure and quantifiable results new and interesting! We do think that some of the conclusions were not as settled as they might have seemed. For example, our motivation in comparing to random sampling is that existing meta-evaluation measures used in prior work do not always show whether micro-benchmarks improve over random sampling, even at extremely small micro-benchmark sizes (e.g., mean estimation error does not show much difference in Figure 3 in [1]). As we discuss in the second paragraph of Section 5.2, MDAD provides additional evidence that micro-benchmarks are better than random sampling at extremely small dataset sizes.
>
> [1]: tinyBenchmarks: evaluating LLMs with fewer examples, Polo et al., ICML 2024.
>
> ***Question:** What is your perspective on how your results echo the message of “the bitter lesson”?*
>
> This is a great question. Our results show the bitter lesson at work: for all micro-benchmarks we consider, increasing evaluation dataset size is the most effective way to more accurately compare models. Micro-benchmarks are useful in extremely compute-limited settings, but if the goal is to accurately rank as many models as possible, then larger evaluation sets are necessary.

---

### Official Review · Reviewer_h3U2 · 2025-11-01

**Soundness:** 4
**Presentation:** 4
**Contribution:** 3
**Rating:** 8
**Confidence:** 3

**Summary:**

The paper presents a comprehensive study on the existing techniques for language model micro-benchmarking, which aims to improve the evaluation efficiency by only selecting a representative subset of the whole benchmark for evaluation. Specifically, this paper introduces a meta-evaluation measure for micro-benchmarking which investigates how well a micro-benchmark can rank two models as a function of their performance difference on the full benchmark. Evaluation shows that no micro-benchmarking method can consistently rank model pairs when the performance difference is small and, when the performance difference becomes larger, random sampling has a comparable performance with other existing micro-benchmarking techniques.

**Strengths:**

- This paper explores an important and interesting research direction.
- The paper is well-written and easy to follow.
- This paper proposes a novel metric, namely MDAD, to evaluate the effectiveness of micro-benchmarking, which is more robust and reliable than existing metrics such as mean
estimation error and rank correlation.
- The evaluation has involved a large number of comprehensive experiments to study the performance and impact of existing micro-benchmarking techniques.

**Weaknesses:**

- While the paper has conducted a very comprehensive evaluation, there’s a lack of concrete case studies in the paper to provide a more intuitive discussion of the results. The readability of the paper can be further improved with more case studies.

**Questions:**

None beyond the above.

---

> ### Author Response · Authors · 2025-11-21
> **Author response to Reviewier h3U2**
>
> Thanks for your review! We appreciate your positive assessment and that you see so many strengths of the paper.
>
> ***Weakness:** Additional case studies would provide a more intuitive discussion of the results and improve the paper’s readability.*
>
> Thanks for this suggestion! We have more clearly delineated existing results from Section 5 as concrete case studies:
> - Section 5: We have focused on BBH as a case study to more concretely show how MDAD decreases with micro-benchmark size (lines 316-323) and how micro-benchmarks have limits at small sizes (lines 348-358).
> - Section 5: We have added a deep dive into the performance stagnation of Anchor Points at larger micro-benchmark sizes (lines 360-373).
> - Section 5.2: We have fleshed out the observation that MDAD can better distinguish micro-benchmark performance from random sampling by focusing on MMLU results (lines 424-435).
> - Section 5.3: We now explicitly note that Section 5.3 is a longer case-study about 8B-parameter instruction-tuned models.

---

### Official Review · Reviewer_3saE · 2025-11-01

**Soundness:** 3
**Presentation:** 3
**Contribution:** 3
**Rating:** 4
**Confidence:** 4

**Summary:**

The paper introduces an evaluation framework for micro-benchmarking. Given a full benchmark $D_{full}$ and a selection strategy *S*, the proposed metric evaluate how faithfully the ranking obtained on the subset $D = S(D_{full}) \subset D_{full}$ reflects the ranking on the full benchmark $D_{full}$. They define the *Minimum Detectable Accuracy Difference* (MDAD), which is the smallest performance gap on the full benchmark for which pairwise model rankings on a micro-benchmark remain consistent with those on the full benchmark $D_{full}$. Their findings are multiple. For example, with a budget of only 10 examples,  none of the selection strategies they consider can correctly rank all pair of models which differs by 3.5 points on MMLU-Pro with at least "80% confidence". Moreover, as the size of the micro-benchmark increases , random selection increasingly becomes competitive with more sophisticated methods.

**Strengths:**

- The paper presents a theoretical approach for evaluating micro-benchmarking. It is important in practice, since assessing large models across multiple full-scale benchmarks can quickly become prohibitively expensive. The proposed approach is valuable in that it provides a way to determine whether given a selection strategy and a budge of $n examples, one can reliably distinguish between models whose performance differs by a specified range `[a, b]`.
- The experiments cover multiple selection strategies, benchmarks and setups which help broadly assessing how their metric behaves.

**Weaknesses:**

- The authors appear to derive MDAD by applying statistical testing principles to micro-benchmarking (somewhat analogous to subset selection) but this is never made explicit. The paper does not clearly define hypotheses (e.g., $H_0$,  $H_1$ ) or discuss error types, leaving the methodological novelty of their approach uncertain. While the formulation itself may be original, the overall idea seems less so.
- Despite the discussion in **Section 5.1**, the benefits of MDAD over metrics such as *mean estimation error* and *rank correlation* remain unclear. Each strategy has its own strengths and weaknesses, but the provided examples fail to convey the broader implications.

**Questions:**

- Why is the **conclusion** merged with the **discussion**?
- Why is **Anchor points** more performing than the other strategies when selecting only few examples ($|D| \leq 100$) and more generally why is it that it better "tells apart" models with a small performance difference better than the other (this advantage disappears as the number of selected examples |$D$| increases)?
- L235: `We use 50 trials, each with a partition of data points and models.` Does it mean you consider 50 seeds and for each seed you partition your data into two sets (one for selection and the other for generalization)?
- Figure 3 (right)/Figure 4: Why does **Anchor points** stagnates (in terms of MDAD) and performs worse than the other strategies when the number of selected samples reaches a certain point (typically $\geq 500$)?
- Line 458-459: What could explain the fact that selecting micro-benchmarks of subtasks independently results in some sort of slight overfitting?
- What is the impact of varying the confidence of the ranking in the MDAD from 80% to say 95%?
- How expensive/time-consuming is it to compute the MDAD?
- Do you think it is possible to adapt the MDAD to buckets of relative/absolute performance difference in terms of scores given as percentage ?

---

> ### Author Response · Authors · 2025-11-21
> **Author response to Reviewer 3saE (part 1 of 2)**
>
> Thanks for your thorough review!
>
> ***Weakness 1:** The paper does not make the connection to statistical testing explicit. The formulation may be original, but the overall idea is less so.*
>
> Our goal is to translate ideas from statistical power analysis to the micro-benchmarking setting (lines 49-52).  As the reviewer points out, statistical power analysis requires choosing an underlying statistical hypothesis test [1]. Such tests often require additional assumptions about independence of data points (which is explicitly violated by micro-benchmark construction), how often two models make the same predictions on examples, or even that model generalization error is low. Hence we chose not to perform a statistical power analysis directly. Instead, we adapt from statistical power analysis the idea of measuring the minimum amount of model performance difference that can be reliably surfaced (at least 80% of the time) by a dataset into a practical, easy-to-compute measure that can be readily interpreted. We want to emphasize that our formulation of MDAD, its application to micro-benchmarking, and our resultant findings with it are novel. Our extended related work section in Appendix A contains this discussion.
>
> [1]: With Little Power Comes Great Responsibility, Card et al, EMNLP 2020.
>
> ***Weakness 2:** The benefits of MDAD over mean estimation error and rank correlation remain unclear.*
>
> MDAD offers three high-level advantages over mean estimation error and Kendall’s tau rank correlation:
>
> *1. MDAD is more fine-grained than Kendall’s tau.* Kendall’s tau rank correlation measures how many pairwise model comparisons are preserved *in the aggregate* across all model comparisons. MDAD instead considers how many pairwise model comparisons are preserved *at different model accuracy differences* and reports the minimum accuracy difference at which at least 80% of model comparisons are preserved.
>
> *2. MDAD is more readily interpretable than Kendall’s tau.* Kendall’s tau values are in the range of -1 to 1. It is hard to interpret what exactly, say, a Kendall’s tau of 0.5 vs. a value of 0.6 means other than that more pairwise model rankings are preserved at higher values. In contrast, the MDAD value is the minimum model performance difference that an evaluation dataset can distinguish at least 80% of the time. An MDAD value of, say, 5 means that the eval dataset can only reliably distinguish models that differ by at least 5 points of accuracy on the full dataset, while an MDAD of 2 means that it can distinguish more similar models that only differ by at least 2 points of accuracy.
>
> *3. Mean estimation error does not say anything about when micro-benchmarks will preserve model rankings. MDAD does.* Because mean estimation error is the average error that a micro-benchmark has in estimating accuracy *for a single model*, it is unable to indicate when model rankings will be preserved by a micro-benchmark. Instead, MDAD explicitly measures how often pairwise model rankings are preserved.
>
> We have updated Section 5.1 to show how its detailed examples support these higher-level takeaways.
>
> ***Question 1:** Why is the conclusion merged with the discussion?*
>
> We have renamed the final section from “Discussion” to “Discussion and Conclusion” to highlight that the conclusion is present.

---

> ### Author Response · Authors · 2025-11-21
> **Author response to Reviewer 3saE (part 2 of 2)**
>
> ***Questions 2/4:** Why does Anchor Points perform better than other methods when selecting very few examples? And why does the performance of Anchor Points stagnate compared to other methods when selecting more examples?*
>
> This is a great question. We suspect that Anchor Points has lower MDAD than other methods when selecting very few examples because its objective is more closely aligned with what MDAD measures. Its method of selecting examples based on correlations in source model confidence produces examples that can better rank models than, say, the way that tinyBenchmarks trains a proxy model based on item response theory, which more indirectly captures model performance correlations.
>
> We hypothesize that the performance stagnation of Anchor Points as more examples are selected is due to very imbalanced cluster sizes in the underlying clustering that it performs. Anchor Points works by first clustering all of the full benchmark’s examples using $k$-medoids and then selecting one example from each cluster. When selecting 10 examples from MMLU-Pro, cluster sizes are relatively even. But when selecting 1,000 examples, there is an extreme size imbalance between the 1,000 clusters of the benchmark examples: 47% of clusters are singletons, and the largest 10% of clusters together contain half of all the examples. Each selected example from the large clusters must stand in for many more data points. Contrast this with tinyBenchmarks, which also performs clustering but uses $k$-means rather than $k$-medoids and uses a different embedding space for examples. In the same setting of selecting 1,000 examples from MMLU with tinyBenchmarks, only 5% of clusters are singletons, and the largest 10% of clusters only contain 21% of examples. At large micro-benchmark sizes, methods other than Anchor Points are better able to take advantage of the additional information provided by more examples. We have added this analysis to Section 5.
>
> ***Question 3:** Does line 235 mean you consider 50 seeds and for each seed you partition your data into two sets (one for selection and the other for generalization)?*
>
> Yes, that is correct. Additionally, we partition the full set of models into source models (to construct the micro-benchmarks) and target models (to meta-evaluate the micro-benchmarks). We have clarified this in the paper.
>
> ***Question 5:** What could explain the fact that selecting micro-benchmarks of subtasks independently results in some sort of slight overfitting?*
>
> We found that model rankings on the original draws of MMLU-Pro subtasks are not as indicative of model rankings on the new draw. We hypothesize that model rankings could be more variable due to the specialized nature of subtasks and the smaller datasets available for measuring these.
>
> ***Question 6:** What is the impact of varying the confidence of the ranking in the MDAD from 80% to say 95%?*
>
> Figure 7 in Appendix C shows the impact of different agreement thresholds (70%, 80%, 90%) on MDAD, and we have now added a 95% threshold to the figure, too. Agreement between a micro-benchmark and the full benchmark is higher for larger differences in model performance (as seen in Figures 1, 2, and 3). This means that as the agreement threshold for MDAD increases, the main effect is that MDAD also increases. All methods tend to experience this MDAD increase, so the overall results for different thresholds are qualitatively similar to what we see with the 80% threshold MDADs used throughout the paper. We have expanded this discussion in Appendix C. We chose 80% as the default threshold in keeping with conventions from statistical power analysis (lines 179-180), but our code flexibly allows for other thresholds.
>
> ***Question 7:** How expensive/time-consuming is it to compute the MDAD?*
>
> Computing MDAD from micro-benchmarking results takes on average 2.40 seconds (with a standard deviation of 0.24 seconds) on a Macbook Pro with an Apple M3 Pro processor across all our experiments. Our results are averaged over 50 micro-benchmarking trials, and the total amount of time to perform 50 trials depends on the method (shown in Appendix D). For example, Anchor Points on MMLU takes on average 6.2 seconds, so computing MDAD over 50 trials of MMLU takes just over 7 minutes from start to finish when including the micro-benchmarking time. We have added this to Section 4 where we discuss MDAD implementation details.
>
> ***Question 8:** Do you think it is possible to adapt the MDAD to buckets of relative/absolute performance difference in terms of scores given as percentage?*
>
> Yes, MDAD can easily be adapted to other metrics, including raw scores or percentages; we have designed it to be flexible about the underlying performance metric. MDAD already reports absolute performance differences, but it could similarly easily operate on relative differences by changing what the buckets are. Thanks for the suggestion!

---

> > ### Comment · Reviewer_3saE · 2025-11-25
> > **Response to the authors**
> >
> > Thank you to the authors for taking the time to address my concerns.
> >
> > - **W1**: While standard statistical power analysis cannot be directly applied to micro-benchmarking, it would still be valuable to clearly articulate which intuitions from statistical testing are being borrowed. Making these connections explicit in the paper or appendix would help interested readers better understand the underlying rationale.
> >
> > - **W2**: Thank you for clarifying the advantages of MDAD, particularly the first two, which are indeed compelling. I would note, however, that the fact that Mean Estimation Error does not reflect pairwise ranking is not inherently a drawback. Score-based and ranking-based formulations serve different purposes depending on the evaluation goal. If I am not mistaken, one can derive an approximate pairwise ranking from accuracy estimates (though imperfectly), whereas the reverse is not generally true. The key issue, then, is how well such a strategy preserves ranking consistency on the full benchmark.
> >
> > - Since MDAD evaluates selection mechanisms and shows Anchor Points performing best, your insights naturally raise the question of whether one could design a selection strategy explicitly or implicitly optimized for MDAD. Thoughts? While this may be beyond the scope of the current paper, I am curious whether the shortcomings of Anchor Points highlighted by MDAD are also reflected in the other two metrics: rank correlation and mean estimation error.
> >
> > - Thank you for the clarifications regarding the seeds.
> >
> > - My request to vary the confidence threshold had several motivations. First, I mentioned 95% to parallel common statistical testing conventions (e.g., 5% or 1% significance). Second, I wanted to understand whether higher confidence levels would require selecting more examples for a given accuracy bucket. Third, by fixing the number of examples, I wished to see whether MDAD increases as confidence increases. Unsurprisingly, the issues observed with Anchor Points (at |$\mathcal{D}$| = 200 or 1000) persist across all confidence thresholds, and the gap appears to widen as the threshold increases.
> >
> >
> > Overall, thank you again for the clarifications. MDAD is interesting, fast and adaptable. After considering your response, I will update my score accordingly.

---

> > > ### Author Response · Authors · 2025-12-01
> > >
> > > Thanks for your response and for your openness to updating your score.
> > >
> > > - W1: We agree. Lines 49-52 explicitly state the intuition from statistical power analysis that we adapt. We have also now added lines 178-180 to explicitly state the connection to statistical power analysis when we define MDAD in Section 3.
> > > - W2: We agree that mean estimation error provides a complementary way of meta-evaluating micro-benchmarks. We believe MDAD is more practically useful because it directly measures the reliability of model comparisons, and comparing models to each other is a primary use-case for micro-benchmarks.
> > > - A data selection strategy optimized for MDAD is a great idea for future work! Please see [our response to Reviewer E4xW](https://openreview.net/forum?id=cReExMQLiK&noteId=e0WAmFizGs) (Question 1) for promising initial results.

---

### Author Response · Authors · 2025-12-02
**Author final remarks**

Our paper introduces a new meta-evaluation measure for language model micro-benchmarking to perform a comprehensive evaluation of common micro-benchmarking methods. We would like to thank the reviewers for their constructive reviews and overall positive assessment. Reviewers found that our proposed measure is valuable for understanding micro-benchmarking (3saE, h3U2, 1D3U, E4xW), noting that the measure provides practical guidance (3saE, E4xW) and overcomes limitations with existing measures to guide future work in this space (3saE, E4xW). They also found the experiments well-designed (1D3U) and comprehensive (h3U2).

Reviewers primarily asked for clarifications, which we have now supplied and incorporated into the paper. Reviewer h3U2 asked for additional concrete results case studies, which we have now added to Section 5 of the paper. Reviewer E4xW asked for clarifications and readability improvements, which we have now made. They also asked whether our proposed measure can be used to guide data selection, for which we have provided promising initial results. **Reviewer 3saE, who gave the lowest initial score of 4, agreed to improve their score after we answered their clarification questions, primarily about error analysis and connections to statistical testing. After the discussion, all reviewers indicated a positive final assessment of the paper.**

---

### Meta-Review · Area_Chair_uoTT · 2025-12-12

**Summary:**

Almost all concerns raised by the reviewers have been adequately addressed. One remaining issue is that, while the reviewers agree the findings are valuable, they do not consider them to be groundbreaking. Therefore, I recommend Accept (oral), but do not support nomination for awards.

**Reviewer Concerns:**

Almost all concerns raised by the reviwers are addressed.

**Reviewer Scores:**

Reviewer 3saE is expected to raise the score to 6, with a small probability of raising it to 8. Reviewer E4xW also has a small probability of raising the score to 8. The remaining reviewers are expected to keep their scores unchanged.

---

### Decision · Program_Chairs · 2026-01-26

Accept (Oral)